# Gaze-dependent evidence accumulation predicts multi-alternative risky choice behaviour

Felix Molter[1,2,3]*, Armin W. Thomas[2,4,5], Scott A. Huettel[6,7], Hauke R. Heekeren[2,8], Peter N. C. Mohr[1,2,3]

1 School of Business & Economics, Freie Universität Berlin, Berlin, Germany, 2 Center for Cognitive Neuroscience, Freie Universität Berlin, Berlin, Germany, 3 WZB Berlin Social Science Center, Berlin, Germany, 4 Department of Electrical Engineering and Computer Science, Technische Universität Berlin, Berlin, Germany, 5 Department of Psychology, Stanford University, Stanford, California, United States of America, 6 Center for Cognitive Neuroscience, Duke University, Durham, North Carolina, United States of America, 7 Department for Psychology and Neuroscience, Duke University, Durham, North Carolina, United States of America, 8 Department for Education and Psychology, Freie Universität Berlin, Berlin, Germany

* felixmolter@gmail.com

**Data Availability Statement:** All raw and preprocessed data, and scripts to reproduce all reported processing, analyses and figures are

## Abstract

Choices are influenced by gaze allocation during deliberation, so that fixating an alternative longer leads to increased probability of choosing it. Gaze-dependent evidence accumulation provides a parsimonious account of choices, response times and gaze-behaviour in many simple decision scenarios. Here, we test whether this framework can also predict more complex context-dependent patterns of choice in a three-alternative risky choice task, where choices and eye movements were subject to attraction and compromise effects. Choices were best described by a gaze-dependent evidence accumulation model, where subjective values of alternatives are discounted while not fixated. Finally, we performed a systematic search over a large model space, allowing us to evaluate the relative contribution of different forms of gaze-dependence and additional mechanisms previously not considered by gaze-dependent accumulation models. Gaze-dependence remained the most important mechanism, but participants with strong attraction effects employed an additional similarity-dependent inhibition mechanism found in other models of multi-alternative multi-attribute choice.

## Author summary

Faced with different choice alternatives, such as food options or risky prospects, our decisions and allocation of gaze (that is where we look) are closely linked, such that items that are looked at longer are often more likely to be chosen. In simple decisions (e.g., choosing between two chocolate bars), these decisions and their associations with gaze allocation are well described by computational models that assume accumulation of evidence in favour of each alternative over time and discounting of momentarily unattended information. However, an important question is whether this class of models can also describe choice behaviour in more complex settings. Specifically, so-called context effects, where

available at https://github.com/moltaire/gda-context.

**Funding:** F.M. was supported by the International Max Planck Research School on the Life Course (LIFE). A.T. was supported by the Max Planck School of Cognition and Stanford Data Science. The funders had no role in study design, data collection and analysis, decision to publish, or preparation of the manuscript.

**Competing interests:** The authors have declared that no competing interests exist.

preferences between two alternatives can vary with the addition of a third alternative, challenge many models of simple decision making. Our study addresses this question by evaluating gaze-dependent evidence accumulation models in a setting where choices between two risky lotteries are systematically influenced by a third alternative. We find gaze-dependent models to be able to describe context effects because decision-makers' gaze allocation also varies with different sets of alternatives.

## Introduction

Imagine you inherited money from a distant relative and you need to decide how to invest it. You reach out to your trusted investment advisor who swiftly sends you a brochure with three different investment alternatives summarized in a single table. You visually inspect the alternatives–comparing their expected returns, associated risks, fees, and other attributes–before your gaze settles on the intermediate option that promises moderate expected returns at a medium level of risk. Why did you choose that option?

Previous work has established that the role of visual attention in decision making under risk exceeds mere information sampling [1–3]. Instead, as in other forms of preferential and perceptual decision making, visual fixations have a constructive role in the decision process, so that alternatives that are looked at for a longer time are generally more likely to be chosen [4–10]. This association between gaze and choice has been formalized in gaze-dependent evidence accumulation models [1,2,5,6,10–13]. The attentional Drift Diffusion Model (aDDM) [6], Gaze-weighted Linear Accumulator Model [12,13], and others that are applied explicitly to risky choice [1] assume that decision makers accumulate evidence in favour of each alternative until evidence for one alternative reaches a threshold and a decision is made. Crucially, accumulation rates are assumed to depend on gaze allocation, so that evidence for an alternative is discounted while it is not fixated. Prior work has tested these assumptions in simple decisions under risk, involving two risky gambles with two equally probable outcomes [3] or two risky gambles described by outcome and probability [1]. Like the example of choosing an investment plan, however, many real-life decisions are more complex than simple binary choice. For example, investment decisions can involve more than two alternatives that vary on multiple attributes (such as expected return, associated risk, and fees), including both sets of options with similarly low expected returns (e.g., government bonds, fixed deposits) and sets of other riskier options promising larger gains at higher risk (e.g., stocks or derivatives).

Crucially, choices in these multi-alternative, multi-attribute settings pose a challenge to many traditional models of risky choice, including Expected Utility Theory (EU) [14] and Prospect Theory [15,16], which obey basic axioms of rational choice like independence of irrelevant alternatives (IIA) [17]. Briefly, IIA asserts that the preference between two alternatives should not depend on other alternatives. At least three context effects–the attraction [18], compromise [19], and similarity [20] effects–show that IIA is frequently violated: The attraction effect describes an increase in preference for an alternative following the addition of a similar, but slightly inferior alternative. The compromise effect describes an increase in preference for an alternative after a third alternative was added that makes it appear as intermediate. The similarity effect predicts that adding a third option that is similar to one of the original options, and equally appealing, will increase relative preference for the other, dissimilar alternative. While these context effects are predominantly investigated using consumer goods, some studies also found them to affect choices between risky gambles [18,20–23]. Generally, models assuming that each alternative can independently be assigned a scalar value denoting

its utility cannot account for context effects [24]. In contrast, models that can predict such effects often assume that preferences are constructed by comparing alternatives on different attribute dimensions [20,25–28] and employ additional psychological or neurobiologically-inspired mechanisms like loss-aversion or inhibition between alternatives.

Few studies have investigated the information search process in context-effect settings to arbitrate between models of context-dependent choice: Noguchi & Stewart found decision makers to shift their gaze mainly between pairs of alternatives on single attribute dimensions [29]. Similarly, Marini *et al.* found that adding an asymmetrically dominated decoy to a choice set shifts gaze towards the target's dominant attribute and leads decision makers to make more transitions between target and decoy alternatives [30].

In contrast to these qualitative uses of eye movement data, the evaluation of behavioural models that formally integrate gaze allocation into the decision process, however, has yet to be performed in context-effect settings. It therefore remains unclear, whether gaze-dependent evidence accumulation models that precisely describe choices, gaze allocation, and their interaction in simple choice, can also account for context-dependent choice behaviour.

Here, we therefore test to what extent gaze-dependent accumulation can explain risky choices in the presence of context effects. Specifically, we focus on attraction and compromise effects, which present critical tests for models of decision making, especially given the makeup of the added decoy alternatives: While being viable choice alternatives in compromise-effect settings, decoys in attraction-effect settings should never be chosen, despite having appetitive character when evaluated on their own.

If the distribution of gaze itself changed depending on the decision context (e.g., shift towards dominant alternatives in the attraction effect, compromise alternatives in the compromise effect), then models that incorporate gaze information could potentially account for violations of IIA–and in turn better predict patterns of risky choices under different contexts. Such attentional effects have been reported for intertemporal choice [30] and for multi-alternative, multi-attribute choice in the absence of context effects [2,31]. Establishing that context effects depend on allocation of gaze (and thus attention) would provide a novel perspective on the mechanisms underlying those effects, linking prior work in behavioural economics and marketing with new models from psychology and neuroscience.

We replicate the attraction and compromise effects in a task where participants made repeated choices in a multi-alternative, multi-attribute setting involving risky gambles. We compare a model of gaze-dependent accumulation adapted from previous work on binary risky choice [1] with an established model of context-dependent multi-alternative, multi-attribute choice without gaze-dependence and with other traditional models of risky choice. Notably, the gaze-dependent accumulation model did not include any dedicated mechanisms to produce context effects. We found the gaze-dependent model to give the overall best account of the choice data, while underestimating strong attraction effects of some individuals. A systematic search over a large model space combining features across model classes confirmed that all participants' behaviour was best described by a gaze-dependent accumulation process, but that individual differences in attraction effect strengths are predicted best by variants integrating mechanisms from other models.

## Results

### Choice task

In our experiment, 40 participants made repeated decisions between three all-or-nothing gambles, each described by a probability $p$ to win an outcome $m$ and nothing otherwise (Fig 1A and 1B). We recorded participants' eye movements during the task with an eye tracker (see

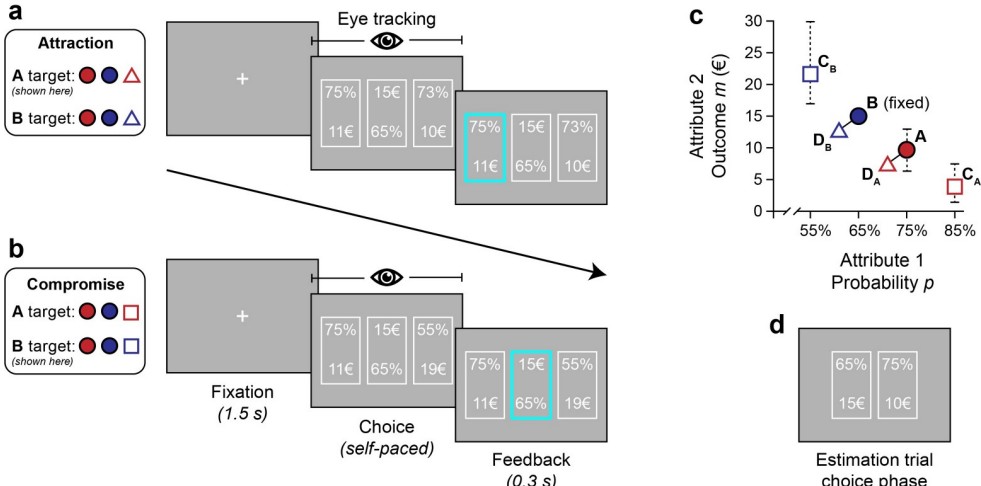

**Fig 1. Choice task and attribute space of stimuli.** Choice task (**a**, **b**). Trials started with a central fixation cross for 1.5 s. Next, three gambles were presented. Participants made a choice by button press, without time limit. Eye movements were recorded during the choice phase. Finally, brief feedback on the choice (but not on the gamble outcome) was displayed. Note that in each trial, the vertical layout of probability and outcome attributes was determined randomly for each alternative. Choice sets in attraction trials (**a**) contained core options $A$ and $B$ and an asymmetrically dominated decoy $D_A$ (shown) or $D_B$. Compromise trials (**b**) included $A$, $B$, and a compromise decoy $C_A$ or $C_B$ (shown). (**c**) Attribute space. Each gamble is described by two attributes: Probability $p$ and outcome $m$. The core options $A$ and $B$ were presented in every trial, along with one of four decoy alternatives: Asymmetrically dominated decoys ($D_A$ or $D_B$) and compromise decoys ($C_A$ or $C_B$) are expected to elicit attraction and compromise effects, respectively. Dashed lines indicate possible stimulus placements. (**d**) Exact positions for $C_B$, $A$ and $C_A$ were calibrated for pairwise indifference in separate binary choice estimation blocks for each individual. Differences in outcome between neighbouring alternatives was not less than 2 EUR. Dominated decoys $D_A$ and $D_B$ were always 2% and 1 EUR worse than $A$ and $B$ respectively.

Methods for details). Participants were instructed that after completing the task, their chosen gamble from one randomly determined trial would be played out for an additional bonus payment. In contrast to hypothetical choices between consumer goods described on attribute dimensions like quality and price, risky gamble stimuli offer a high amount of control over their attributes and a straightforward way to incentivise choices. The gamble stimuli were designed to elicit attraction and compromise effects and were individually tailored to account for each participant's risk preferences (Fig 1C and 1D).

Focusing on attraction and compromise effects allowed us to include a high number of trials per effect, without exceeding a total experiment duration of 90 minutes for most participants. In addition, attraction and compromise effects differ in the expected role of the third alternative on choice, providing an additional challenge to decision models: In the attraction effect, dominated decoys are typically not chosen, whereas extreme decoys provide viable choice alternatives in compromise settings. Furthermore, compromise-making decoys lie at the extremes of the attribute space, introducing additional variability in attribute values experienced by participants, and providing an additional challenge for behavioural models of decision making.

Participants performed three pairs of indifference-estimation and experimental blocks, for a total of 225 experimental trials (see Methods for additional details on the experimental procedure and gamble stimuli). In indifference-estimation blocks, participants made repeated choices between pairs of gambles with different probabilities (Fig 1D). Gamble outcomes $m$ were adjusted according to participants' choices so that four approximately equally preferred gambles $C_B$, $B$, $A$, and $C_A$ were constructed with winning probabilities $p$ = 55%, 65%, 75%, and

85%. Additionally, two asymmetrically dominated decoys $D_A$ and $D_B$ were defined to be 2% and 1 EUR worse than gambles $A$ and $B$, respectively. Following this indifference estimation, participants performed an experimental block of 75 ternary choice trials, 32 of which were compromise trials balanced with respect to the target option (i.e., 16 choice sets $\{A, B, C_A\}$, 16 choice sets $\{A, B, C_B\}$), 32 attraction trials (16 choice sets $\{A, B, D_A\}$, 16 choice sets $\{A, B, D_B\}$) and 11 distractor trials showing randomly created options with expected value of 10 EUR and low (16–32%), medium (37–67%) and high (72–89%) probability $p$. To reduce repetition of stimulus values, noise was added to all outcomes $m$ (-1 EUR or no change) and probabilities $p$ (–3% or +3%) in each trial. Asymmetrically dominated decoys $D_A$ and $D_B$ received the same noise as the target option to preserve the dominance relation.

## Context effects

We first analysed participants' choice behaviour to test whether their choices could be influenced by the set of available alternatives (see Table 1 and Fig 2). Participants rarely chose the dominated decoys in attraction trials (mean ± s.d. = 0.02 ± 0.02 of attraction trials), indicating that participants understood the dominance relationships among the stimuli. In compromise trials, participants chose (non-dominated) decoy alternatives more frequently (mean ± s.d. = 0.27 ± 0.16). In particular, the decoy with the highest possible outcome $C_B$ ($p \approx 55\%$, $m \geq 18$ EUR) was chosen most frequently when it was available. Note that decoy choices in compromise trials are expected, as extreme decoys were specifically calibrated to be approximately equally preferred to the core options.

We tested the presence of the attraction and compromise effects by first computing the relative choice share of the target alternative (RST) [32] for each individual, and separately for attraction and compromise trials. The RST is computed as

$$RST = \frac{N_{\text{target}}}{N_{\text{target}} + N_{\text{competitor}}} \tag{1}$$

where $N$ is the number of times an alternative was chosen. Given the balanced design, where each core alternative acts as a target and competitor equally often, the RST should be close to 0.5 if no context effects are present. If, however, the RST is different from 0.5, a systematic context effect is indicated. We tested whether the mean RST significantly differed from 0.5 across participants by computing its 95% highest posterior density interval (HDI$_{95}$) using Bayesian estimation (BEST) [33,34].

We find evidence for the attraction effect: The mean RST in attraction trials differed meaningfully from 0.5 (mean RST = 0.56, HDI$_{95}$ = [0.51, 0.60], mean $d$ = 0.49, HDI$_{95}$ = [0.10, 0.80]). 25 of 40 (62.5%) participants had an RST above 0.5 in attraction trials. Notably, similar to previous work [35] a subgroup of participants (9 of 40, 22.5%) showed particularly strong attraction effects with individual RSTs above 0.7. We could not, however, find evidence that these

**Table 1. Relative choice frequencies across participants in the four trial types.** Across the group, target options were chosen more frequently than competitors in both types of attraction ($D_A$ and $D_B$) and compromise trials ($C_A$, $C_B$). Dominated decoys were almost never chosen. In compromise trials including the high-outcome decoy $C_B$, the decoy was chosen more frequently than core options.

| Trial type | Target | Competitor | Decoy |
|---|---|---|---|
| $D_A$ | 0.60 | 0.39 | 0.01 |
| $D_B$ | 0.53 | 0.45 | 0.02 |
| $C_A$ | 0.46 | 0.37 | 0.16 |
| $C_B$ | 0.32 | 0.31 | 0.37 |

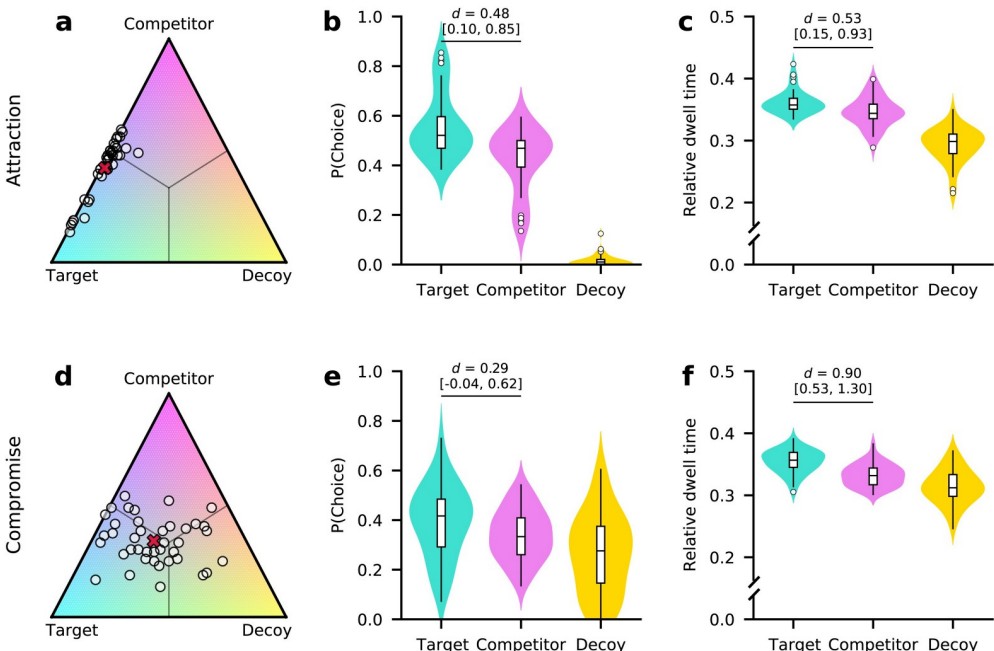

**Fig 2. Context effects are present in choices and relative dwell time.** Participants' choices were influenced by asymmetrically dominated decoys and, to a lesser extent, by extreme compromise-making decoys. **(a, d)** Ternary plots of individual relative choice frequencies for target (lower left, teal), competitor (top, pink), and decoy (lower right, yellow) alternatives in attraction **(a)** and compromise **(d)** trials. Each dot represents one participant. The position on the simplex indicates relative choice frequencies for alternatives. Straight lines from the centre indicate equal frequencies for two alternatives. The red "x" indicates the group average. **(b, e)** Relative choice frequencies in attraction **(b)** and compromise **(e)** trials. In attraction trials, some participants strongly favoured the target alternative and almost no decoy choices were made. While target alternatives are still chosen more frequently than competitors in compromise trials, the effect is less pronounced, and extreme decoys are still chosen frequently. **(c, f)** Relative dwell time towards alternatives. In both, attraction **(c)** and compromise **(f)** trials, target alternatives received greater relative dwell times than competitors. $d$ denotes Cohen's $d$ from paired BEST analysis with HDI$_{95}$ given in brackets. Violin plots show kernel density estimates of distributions of individual values. Box plots mark lower and upper quartiles and median. Whiskers extend from first and last datum within 1.5 times the interquartile range from lower and upper quartiles, respectively. Values outside this range are indicated by open circles.

individuals used the dominance relationship as a simplifying choice rule (see S3 Note). To allow comparisons with other studies that quantified context effects as the difference between choice shares between targets and competitors, we also report these differences: In attraction trials, the average difference was 0.12 (HDI$_{95}$ = [0.01, 0.20], mean $d$ = 0.48, HDI$_{95}$ = [0.10, 0.85], Fig 2B).

We only found weak evidence for the compromise effect using the gamble stimuli: The mean RST in compromise trials was 0.53, but its estimated HDI$_{95}$ included 0.5 (HDI$_{95}$ = [0.49, 0.57], 91.1% of posterior mass above 0.5, mean $d$ = 0.23, HDI$_{95}$ = [-0.11, 0.59]). 26 of 40 (65%) participants showed an RST above 0.5 in compromise trials. The mean difference between choice shares for targets and competitors was 0.05 (HDI$_{95}$ = [-0.01, 0.11], mean $d$ = 0.29, HDI$_{95}$ = [-0.04, 0.62], 95.8% of posterior mass above 0; Fig 2E). These results are similar to the marginal effects obtained using perceptual stimuli in previous work [36,37].

Of note, compromise effects were stronger in trials with target alternative $A$ ($p \approx 75\%$, $m \approx 10$ EUR) than in those with target alternative $B$ ($p \approx 65\%$, $m \approx 15$ EUR; Table 1). A related asymmetry is present in the proportion of decoy choices in these trials. One possible explanation for these asymmetries comes from the indifference estimation procedure ran before each block: Here, the outcome amounts of alternatives $A$, $C_A$, and $C_B$ were estimated so that

alternatives were roughly equally preferred. The fidelity of this procedure, however, is limited by at least two factors: First, the outcome amount of the $C_A$ alternative was adapted in relation to the $A$ alternative, whose amount was estimated itself. Therefore, alternative $C_A$'s estimated amount might be associated with higher estimation error. Second, the range of possible amounts for the $C_A$ alternative was limited by alternative $A$ and a minimum outcome of 1 EUR. Finally, it remains possible, that participants' preferences shifted towards higher outcomes after the estimation block, leading to stronger preferences for the $C_B$ decoy and reduced compromise effects in this condition. However, preference for higher outcomes was only observed in compromise and attraction trials targeting alternative $B$ (Table 1). Other alternatives were preferred in the remaining conditions, and overall participants chose the highest outcome only in 38% of the trials (in distractor trials, the highest outcome was only chosen in 21% of trials).

Similar to other work [26,32], we found a positive correlation between the effects across participants, even though it was weaker in our task ($r = 0.24$, $HDI_{95} = [-0.05, 0.51]$, 94.5% of posterior mass above 0). In contrast to previous work [2,25,38–40], context effect strengths were not related to mean response times across participants (S1 Fig). On the individual level, participants were not significantly more likely to choose the target in attraction trials with response times longer than their median response time compared to faster trials (mean difference = 2.7%, $HDI_{95} = [-2.0, 7.5\%]$, mean $d = 0.19$, $HDI_{95} = [-0.12, 0.511]$). There was, however, a marginal effect in compromise trials, so that participants were more likely to choose the target in trials with slower response times (mean difference = 3.8%, $HDI_{95} = [0.0, 7.5\%]$, mean $d = 0.33$, $HDI_{95} = [-0.004, 0.63]$; 97.90% of posterior mass above 0).

Taken together, we successfully induced context effects within our participant sample, with non-trivial variability in the strength of those effects across individuals. These data provide a complex testing scenario to investigate gaze-bias effects in multi-alternative multi-attribute choice and to compare gaze-dependent accumulation models with competing theories.

## Context effects are present in relative dwell times

Next, we tested whether participants' eye movements were affected by the set of available alternatives, similar to their choices. We therefore computed each alternative's relative dwell time in a trial by summing the duration of all fixations durations towards it, and normalizing to the sum of all fixations in the trial. Both patterns of context effects of choice behaviour were present in the relative dwell data: Target options received greater relative dwell time than competitors in attraction (mean difference = 0.014, $HDI_{95} = [0.004, 0.024]$, mean $d = 0.53$, $HDI_{95} = [0.15, 0.93]$, Fig 2C) and compromise trials (mean difference = 0.022, $HDI_{95} = [0.014, 0.030]$, mean $d = 0.90$, $HDI_{95} = [0.53, 1.30]$, Fig 2F). Targets and competitors also received greater relative dwell time than decoys in both trial types.

Comparing attraction and compromise trials, we found target alternatives to be looked at marginally longer in attraction trials (mean difference = 0.006, $HDI_{95} = [-0.01, 0.015]$, mean $d = 0.27$, $HDI_{95} = [-0.06, 0.59]$, 94.85% of posterior mass above 0). Competitors were also looked at longer in attraction trials (mean difference = 0.013, $HDI_{95} = [0.005, 0.023]$, mean $d = 0.52$, $HDI_{95} = [0.19, 0.89]$), whereas decoys were looked at longer in compromise trials (mean difference = 0.020, $HDI_{95} = [0.007, 0.033]$, mean $d = 0.53$, $HDI_{95} = [0.20, 0.90]$).

To better understand participants' eye movements over the course of the trial, we performed additional exploratory analyses of gaze allocation and transitions. We found an increasing association between gaze and choice over the trial, and longer gaze towards targets, even accounting for choice. Information search occurred both within and between alternatives, with slightly more transitions within alternatives. In addition, we analysed the number of

transitions between target, competitor, and decoy alternatives. Here, we could replicate a recent finding [30] of more transitions between targets and decoys than between competitors and decoys in both attraction and compromise trials. We refer to S1 Note for additional details on these analyses.

We performed multiple additional analyses of the relationship between participants' gaze to decoy alternatives and their individual context effect strengths: First, on a between-participant level, we tested whether strong context effects could be associated with greater relative dwell times towards decoys. There was a meaningful positive relationship between individual mean dwell time towards decoys and RST in attraction trials (mean $r$ = 0.38, $HDI_{95}$ = [0.11, 0.65]), but not compromise trials, where we found a trend of a negative association (mean $r$ = -0.24, $HDI_{95}$ = [-0.52, 0.03]; S5 Fig). Next, we analysed whether these effects were also present on the level of the individual. We therefore split each individual's choice data into trials with shorter and longer relative dwells to the decoy (based on individuals' median dwell time to decoys), separately for attraction and compromise trials. Finally, we performed BEST analyses of individuals' probability of choosing the target alternative in trials with longer vs. shorter dwell towards the decoy. In attraction trials, participants showed 11.6% more target choices for longer compared to shorter dwell times towards the decoy ($HDI_{95}$ = [9.3%, 14.1%]; mean $d$ = 1.78, $HDI_{95}$ = [1.18, 2.45]). In contrast, greater dwell times towards the decoy were associated with a 10.7% decrease in target choices ($HDI_{95}$ = [-15.2%, -6.0%], mean $d$ = -0.82, $HDI_{95}$ = [-1.26, -0.38]) in compromise trials.

## Behavioural modelling and model comparison

Given that participants' choices and eye movements exhibited modulation by the context of available options, we next tested whether their behaviour could be described using a gaze-dependent accumulation model, and how such a model performs in comparison to extant theories of multi-alternative multi-attribute choice. To this end, we adapted a recently developed, gaze-dependent leaky accumulator model of two-alternative risky choice [1] to the three-alternative scenario. We refer to this model as the Gaze-dependent Leaky Accumulator (GLA). It assumes that subjective utilities for each alternative $i$ are constructed as in Cumulative Prospect Theory [16] by first applying a probability weighting function, that transforms objective probabilities $p$ into subjective decision weights. Outcomes $m$ are transformed into utilities using a power function. Next, this model assumes that for each alternative $i$ subjective utilities $x_i^{GLA}$ are accumulated with leakage over the time course of each trial and that utilities are discounted while they are not fixated. At each fixation $n$, accumulators evolve according to

$$X_i^{GLA}(n) = \begin{cases} (1-\lambda) \cdot X_i^{GLA}(n-1) + 1 \cdot x_i^{GLA} & \text{if } i \text{ fixated} \\ (1-\lambda) \cdot X_i^{GLA}(n-1) + \theta \cdot x_i^{GLA} & \text{otherwise} \end{cases} \tag{2}$$

where all $X_i(0)$ = 0. Similar to the aDDM, the $\theta$ parameter ($0 \leq \theta \leq 1$) controls discounting of unattended alternative utilities. The $\lambda$ parameter ($0 \leq \lambda \leq 1$) controls the strength of the accumulation leak. Even though GLA's leak term does not depend on gaze allocation itself, but is constant across alternatives and over time, it allows the model to also use choice-relevant information contained in the temporal order of fixations, in combination with the gaze-discount (see S4 Fig and S2 Note for details). We computed choice probabilities from this model by applying the soft-max choice rule (Eq (5)) over the final accumulator values $X_i^{GLA}$.

Additionally, we fitted an implementation of Multialternative Decision Field Theory (MDFT) [25], representing the class of dynamic multi-attribute, multi-alternative choice theories that have been designed to explain context effects. In contrast to GLA, MDFT assumes that preferences evolve dynamically over time by accumulation of attribute-wise comparisons

between alternatives, and alternatives inhibit each other depending on their distance (or similarity). MDFT does not assume any influence of gaze on the decision process. We also included three baseline models in our comparison. First, we included Prospect Theory, which assumes the same alternative-wise valuation as GLA, but is agnostic to gaze data. Note that for the all-or-nothing gambles used in our experiment, predictions of Prospect Theory and Cumulative Prospect Theory are identical. Second, a static gaze baseline model $GB_{stat}$ predicted choices only from trial-aggregated gaze towards each alternative, ignoring outcome values and outcome probabilities. Third, we constructed a dynamic gaze baseline model $GB_{dyn}$ that, like GLA, assumed leaky evidence accumulation over fixations, but ignored attribute values, using only the sequence of fixations in each trial to predict choices. See Methods for details on these models' implementation and parameter estimation.

All models were fit individually to the choice data of each participant. We compared the models using the Bayesian Information Criterion (BIC) [41], which penalizes more complex models and therefore accounts for differences in complexity between models. To validate our modelling analyses, we performed parameter- and model-recovery analyses. Most models' parameters could be recovered well (S6 Fig). Similarly, data-generating models could be identified (S7 and S8 Figs).

Across the group, all models performed better than the random baseline model (Fig 3A). The GLA had the lowest BIC (mean ± s.d. = 230.63 ± 66.80), followed by the dynamic gaze baseline model $GB_{dyn}$ (mean ± s.d. = 302.46 ± 86.15), MDFT (mean ± s.d. = 345.77 ± 82.44), PT (mean ± s.d. = 355.26 ± 69.57), and the static gaze baseline $GB_{stat}$ (mean ± s.d. = 414.81 ± 36.81). Simulating choices from the models using maximum-likelihood estimates, the proportion of correctly predicted choices was 74.3% for GLA, 62.9% for $GB_{dyn}$, 57.9% for MDFT, 55.3% for PT, and 45.6% for $GB_{stat}$ (see S11 Fig for distributions of model-predicted choice probabilities). Note, that target and competitor options (and decoys in compromise trials) were designed to be closely matched in value, resulting in trials with high choice difficulty, limiting overall model performance. On the individual level, based on lowest BIC scores, the majority (36 of 40, 90%) of participants were best described by GLA. Three participants (7.5%) were best described by MDFT, and one by the dynamic gaze-baseline model (Fig 3B). Protected exceedance probabilities [42], which measure the likelihood that model is more frequent than all others, unambiguously identified GLA as the most likely model ($pXP_{GLA}$ = 1, Fig 3B inset).

Note that, as in previous work [1], the GLA performed better using a temporal resolution of whole fixations, than a variant that made 10 ms steps and therefore used the individual fixations' durations (29 of 40 participants were better described by the first variant; mean ± s.d. BIC $GLA_{dur.}$ = 240.62 ± 65.42). Conclusions of the comparison with the other models, however, do not differ between variants (S9 Fig).

The estimates for GLA's gaze-discount parameter $\theta$, which describes the degree to which alternative's values are attenuated while not fixated, indicate that participants exhibited a moderate gaze-discount on average: $\theta$ estimates ranged from 0.13 (strong gaze-discount) to 0.95 (almost no gaze-discount), with mean ± s.d. = 0.69 ± 0.18. Estimates for the leak parameter ranged from 0.08 (almost no leak) to 0.65 (moderately strong leak), with mean ± s.d. = 0.29 ± 0.20. Individual parameter estimates of all GLA parameters are plotted in S10 Fig and summarised in S1 Table. Notably, both parameters' estimates could be recovered with high accuracy and no bias (S8C and S8D Fig).

Given that MDFT outperforms utility-based models when choices are influenced by context [32], we tested whether model fit of MDFT (relative to GLA) was higher for participants with higher RST. In other words: Does MDFT perform better for stronger context effects? Overall, 7 out of 9 participants with an RST above 0.7 in attraction trials were clearly best described by

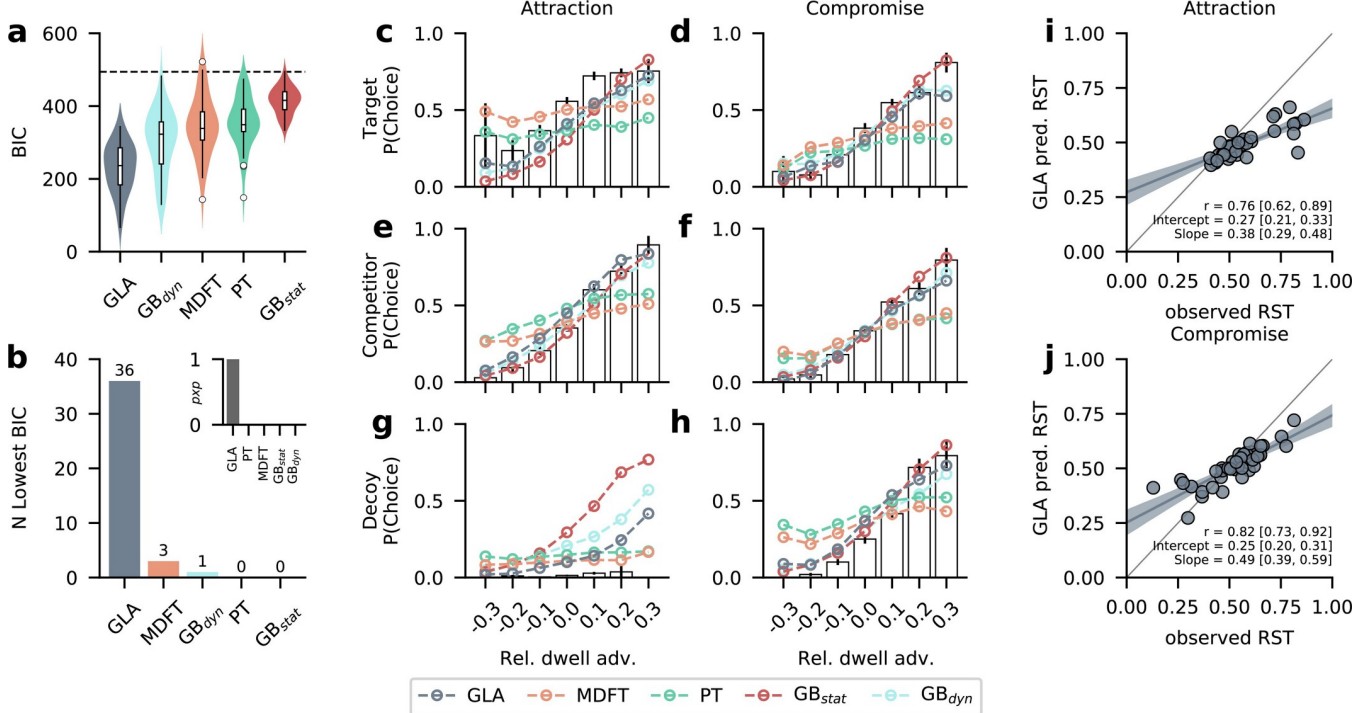

**Fig 3. Model comparison and predictions. (a)** The gaze-dependent leaky accumulator (GLA) provided the best fit (lowest mean BIC) across participants, followed by the dynamic gaze baseline model, MDFT, PT, and the static gaze baseline model. The dashed line indicates the BIC of the random choice baseline model. Violin plots show kernel density estimates of distributions of individual values. Box plots mark lower and upper quartiles and median. Whiskers extend from first and last datum within 1.5 times the interquartile range from lower and upper quartiles, respectively. Values outside this range are indicated by open circles. **(b)** The GLA fitted most (36 of 40, 90%) participants best, with a protected exceedance probability of 1 (inset). **(c-h)** Observed and model-predicted probability of choosing the target **(c, d)**, competitor **(e, f)**, or decoy **(g, h)** alternatives in attraction **(c, e, g)** and compromise trials **(d, f, h)** as a function of relative dwell time advantage. Relative dwell time advantage was computed as relative dwell time towards an alternative minus the mean relative dwell time to all other alternatives. White bars and error bars show mean ± s.e. observed data from even-numbered trials. Model predictions (coloured lines) are based on 50 simulations of each odd-numbered trial. **(i, j)** Observed and predicted RST of the best-fitting GLA for attraction **(i)** and compromise **(j)** trials. Each circle represents one participant. The winning model's predicted context effect sizes correlated significantly with the observed ones. Strong context effects, however, were underestimated, as indicated by the reduced slopes. GLA: Gaze-dependent Leaky Accumulator. MDFT: Multialternative Decision Field Theory. PT: Prospect Theory. GB$_{stat}$: Static Gaze Baseline. GB$_{dyn}$: Dynamic Gaze Baseline.

GLA. While we found that the relative superiority of GLA over MDFT decreased with strong attraction effects, this was not the case for compromise trials (S12 Fig), suggesting that some features of MDFT might help capture strong attraction effects.

As an additional indicator of model fit, we tested whether the models could quantitatively reproduce the observed positive association of gaze and choice (S1D, S1J and S2 Figs). Specifically, following previous work [6,11], we inspected the model-predicted probability of choosing an alternative as a function of its relative dwell time advantage: the difference in relative dwell time towards an alternative minus the mean relative dwell time to other alternatives. The probability of choosing an alternative generally increased with its relative dwell time advantage (Fig 3C–3H), except for dominated decoys in attraction trials, which were not chosen even if looked at longer than other alternatives (Fig 3G).

All gaze-dependent models were able to capture this positive association. Note, however, that they also predicted choices of dominated decoys in attraction trials, if decoys had a large dwell time advantage (Fig 3G). In this case, GLA performed better than GB$_{dyn}$ and GB$_{stat}$, as it predicted fewer decoy choices. However, MDFT and PT generally could not capture the empirical association of gaze and choice fully; they predicted too many choices of alternatives

with negative dwell time advantage, and too few choices of alternatives with positive dwell time advantage. Nor did they predict choices of dominated decoys, even if the decoy was looked at longer. Yet, MDFT and PT still predicted a weak positive association of gaze and choice (Fig 3C–3F) that probably results from participants looking longer at items they value highly [2,43] and eventually choose. When gaze-independent models like MDFT and PT then correctly predict choice, they indirectly also predict an association with gaze, even though they underestimate the strength of this association considerably.

Finally, we investigated the ability of the best-performing model to predict individual differences in context effect strengths. Therefore, we predicted choices from the fitted GLA model and correlated the resulting RST with the observed data. Predicted RST correlated significantly with observed ones in attraction ($r = 0.76$, $HDI_{95} = [0.62, 0.90]$, Fig 3I) and compromise ($r = 0.82$, $HDI_{95} = [0.72, 0.92]$, Fig 3J) trials. However, the model underestimates large deviations from RST = 0.5, suggesting that its gaze-discount mechanism can capture the qualitative pattern of context effects but not their expression in participants with extreme RSTs.

Inspection of predicted choice probabilities (S11 Fig) shows that, on average, GLA predicted high probabilities for the empirically chosen alternative (indicating good overall fit), but comparable proportions of target and competitor choices (resulting in reduced RST). Other models, like MDFT, predicted larger differences between target and competitor choices for some participants, but assigned lower probability to empirically chosen alternatives (resulting in overall inferior fit)

We note, however, that the presented comparison of extant models is unbalanced with respect to the different theories' use (or non-use) of the eye tracking data. Given the generally positive association between gaze and choice [13], this puts models without gaze-dependence at a disadvantage in this comparison. Similarly, models' inability to quantitatively capture associations between choice and gaze data, is a direct consequence of their gaze-*in*dependence. This imbalance with respect to the use of gaze-data, as well as the overall large differences between the competing models' architectures, make it difficult to attribute differences in model performance to individual model mechanisms. For example, some MDFT mechanism might be associated with higher model performance, but its advantage is "washed out" by the disadvantage of MDFT's lack of gaze-dependence. To perform a more balanced comparison of models, and allow evaluation of individual model parameters, we performed a more exhaustive and systematic model comparison, described in the following.

## Systematic exploration of a large model space

The model comparison identified the advantage of the gaze-dependent accumulation model over its competitors. To better understand the contribution of individual model mechanisms (such as accumulation leak, inhibition between alternatives, or the gaze-dependent discount) to model performance, we performed a search across a large, systematically designed model space, in a so-called "switchboard analysis" [44]. Here, the idea is to isolate, group and exhaustively combine mechanistic assumptions of different models. Each group of mechanisms is considered a switch that can take different levels (e.g., an "inhibition" switch could take the levels "distance-dependent" as in MDFT, "constant" or "none"). Ultimately, this approach can be used to gauge the contribution of individual model mechanisms (opposed to evaluating whole models or model classes as in the more traditional model comparison presented above). In addition, it provides a systematic way to generate novel hybrid models, combining mechanisms from different *a priori* defined models.

This analysis approach further allowed us to test different assumptions about the functional forms of the gaze bias mechanism (e.g., as discounting non-fixated alternatives' values,

controlling accumulation leak, among others [45]). We therefore expanded the range of gaze-dependent mechanisms from the original set of models to include additional eye-movement related switches and levels, like attribute- and alternative-wise gaze-discounts, gaze-dependent leakage and inhibition. This allowed us to test if gaze-bias implementations differed from the ones usually used in gaze-dependent accumulation [3,6,10–13,46,47].

Our switchboard comprised a total of 192 model variants, derived from six switches that were combined exhaustively: Attribute integration (additive vs. multiplicative), evidence comparison (independent accumulation for each alternative or comparative accumulation of evidence relative to other alternatives), alternative-wise and attribute-wise gaze-discount (included or not), accumulation leak (none, constant or gaze-dependent) and inhibition between alternatives (none, constant, gaze-dependent or distance-dependent). The models generally resembled the form of the *a priori* defined GLA, but with substantial differences depending on the specific set of switch levels (Fig 4A; see Methods and S2 Table for details on the framework and switch levels). As before, each model variant was fit to the individual data of each participant and model performance was evaluated based on the BIC score. Note that the GLA coincides with one of the 192 variants (variant A in Table 2; multiplicative attribute integration, alternative-wise gaze-discount, constant leak, no inhibition, no comparison). Similarly, one variant (not in Table 2) conceptually resembles MDFT in some, but not all aspects (additive attribute integration, comparative evidence accumulation, constant leak, distance-based inhibition, strong attribute-wise gaze-discount).

**Best model mechanisms.**   On the level of model mechanisms, multiplicative attribute integration outperformed additive integration (mean $BIC_{mul.} = 312.25$, mean $BIC_{add.} = 333.20$; Fig 4B), inclusion of an alternative-wise gaze-discount (mean $BIC_{GD\ alt.} = 289.13$, mean $BIC_{no\ GD\ alt.} = 356.32$), but not attribute-wise gaze-discount yielded lower BIC scores (mean $BIC_{GD\ att.} = 324.11$, mean $BIC_{no\ GD\ att.} = 321.33$). Other gaze-dependent mechanisms also improved model fit: Variants with gaze-dependent leak yielded lower BIC scores than variants with constant or no leak ($BIC_{gaze} = 292.06$, $BIC_{constant} = 304.84$, $BIC_{none} = 381.49$). Gaze-dependent inhibition performed better than constant, none or distance-dependent inhibition ($BIC_{gaze} = 305.64$, $BIC_{constant} = 316.06$, $BIC_{none} = 311.75$, $BIC_{distance} = 351.57$). In summary, all mechanisms that allowed a model to exploit the positive association between gaze and choice improved model fit on average.

Counting switch-values of individually best fitting variants, most participants were best described by model variants with multiplicative integration, with alternative-wise and no attribute-wise gaze-discount, with constant accumulation leak parameter and no inhibition (S13 Fig)

**Best model variants.**   The overall best-fitting model variant was the variant identical to the GLA (Fig 4C, S3 Table): It included multiplicative attribute integration, an alternative-wise gaze-discount, and constant leak. Note that our analysis did not allow us to distinguish independent and comparative accumulation for this variant, because they yield equivalent results. All the ten best performing models used multiplicative attribute integration, and most used an alternative-wise but no attribute-wise gaze-discount and constant leak (S3 Table). Results were more ambiguous for the comparison and inhibition mechanisms.

Notably, one of the best ten variants employed a distance-dependent inhibition mechanism, and used other features resembling MDFT like constant leak, and accumulation of comparative values. Unlike MDFT, but like GLA, this variant used an alternative-wise gaze-discount, no attribute-wise mechanism of attention, and multiplicative integration of attributes. While not achieving the best overall fit, this hybrid variant performed significantly better than the original MDFT implementation (mean $BIC_{MDFT} = 345.77$; Fig 3A).

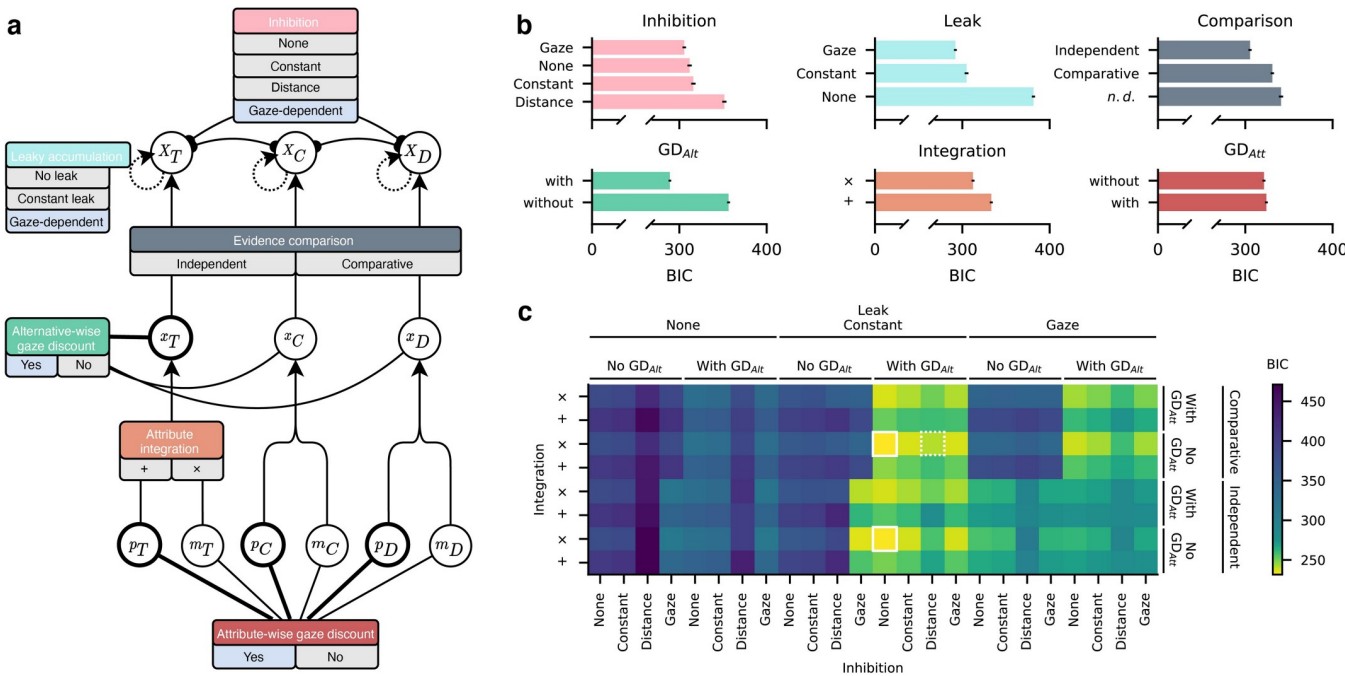

**Fig 4. Switchboard analysis. (a)** Overview of general switchboard framework and individual switches from which individual variants are constructed by setting the switches to a set of levels. Gaze-dependent switch levels are shaded blue. Attributes can be discounted based on gaze (lower level) and are integrated into alternative values (middle level). Alternative values can be discounted based on gaze, compared, and integrated (upper level) with leak and inhibition. **(b)** Average model fit associated with each switch's levels. Each bar shows the average BIC for all model variants that had the respective switch set to this level (e.g., first panel, top bar: average BIC of all variants with gaze-dependent inhibition). Gaze-dependent inhibition and leak, independent evidence accumulation, alternative-wise gaze-discount, multiplicative attribute integration, and no attribute-wise gaze-discount yielded lower BIC on average. **(c)** Overview of mean BIC for each of 192 model variants. More yellow colours indicate lower BIC and better model fit. The variant with the lowest BIC is identical to the GLA (alternative-wise, no attribute-wise gaze-discount, multiplicative attribute integration, constant leak, and no inhibition) and is outlined in white. The hybrid variant which described 9 participants, mostly with strong attraction effects, best (see main text and Fig 5) has a dotted outline. Note that some variants were mathematically equivalent (see main text and Methods) including the variant with lowest BIC, which is therefore highlighted twice.

The GLA variant also described 17 of 40 (42.5%) participants best (Table 2). Another similar variant with additive attribute integration described an additional four participants best. An additive relationship between attributes is typically assumed by models of multi-attribute multi-alternative choice [25,27]. Furthermore, additive integration of probability and outcome has recently been suggested as an alternative to multiplicative mechanisms and has been demonstrated to be equivalent for particular parameterisations and parameter values [48]. Five participants were best described by variants similar to the GLA, but using gaze-dependent leakage or inhibition instead of an alternative-wise gaze-discount. Note that gaze-dependent inhibition and leakage mechanisms can act similarly to an alternative-wise gaze-discount: All three mechanisms effectively reduce accumulated evidence of non-fixated alternatives. While the alternative-wise gaze bias mechanism applies a constant discount to the accumulation inputs, the gaze-dependent inhibition mechanism is proportional to the accumulated evidence of the currently fixated alternative, and applies to the already accumulated evidence of other options, not the inputs to the accumulation process. Gaze-dependent leakage similarly reduces already accumulated evidence, proportional to the momentary accumulator value.

Notably, nine participants (22.5%) were best described by the previously described hybrid variant using a distance-dependent inhibition mechanism (Table 2). Additional two participants were best described by other variants using distance-dependent inhibition in conjunction with an alternative-wise gaze-discount.

**Hybrid variant.**   Finally, we analysed the hybrid model variant in more detail (variant B in Table 2), which described 9 participants best. This variant performed especially well for participants with large attraction effects (Fig 5A), whereas GLA best described most participants with attraction RST around 0.5. In contrast, better-performing variants could not be separated by compromise effect strength (Fig 5B). The hybrid variant correctly predicted individual differences in the attraction effect (correlation between observed and predicted RST: $r = 0.92$, $HDI_{95} = [0.86\ 0.96]$; Fig 5C). Here, it performed better than the GLA model (Fig 3I), as it was also able to capture the behaviour of participants with particularly strong attraction effects: Using distance-dependent inhibition, it was able to predict high choice probabilities for target alternatives in attraction trials for some participants, and fewer choices of competitor and dominated decoy alternatives (S11G Fig). Predictions of individual RST in compromise trials were almost identical between the two models ($r = 0.81$, $HDI_{95} = [0.70, 0.91]$; see Figs 5D, 3J for GLA).

**Table 2. Overview of individually best fitting model variants.** *N* indicates the number of participants best described by the variant described in the row. The top variant (A) coincided with the GLA model. Note that all individually best fitting models had some form of gaze-dependence (blue shaded cells, mostly alternative-wise gaze-discount). *"n.d."* denotes variants where comparison mechanisms were not distinguishable by the analysis. The last two columns show observed mean (± s.d. if applicable) RST of participants best described by each variant, for attraction and compromise trials, respectively.

|   | N | GD_Alt | GD_Att | Leak | Inhibition | Integration | Comparison | Attr. RST | Comp. RST |
|---|---|--------|--------|------|------------|-------------|------------|-----------|-----------|
| A | 17 | Yes | No | Constant | None | Multiplicative | *n.d.* | 0.51 ± 0.07 | 0.55 ± 0.08 |
| B | 9 | Yes | No | Constant | Distance | Multiplicative | Comparative | 0.75 ± 0.12 | 0.56 ± 0.16 |
| C | 4 | Yes | No | Constant | None | Additive | *n.d.* | 0.52 ± 0.05 | 0.42 ± 0.14 |
| D | 3 | No | No | Gaze | None | Multiplicative | Independent | 0.59 ± 0.02 | 0.54 ± 0.11 |
| E | 2 | No | No | Constant | Gaze | Multiplicative | Independent | 0.50 ± 0.00 | 0.51 ± 0.02 |
| F | 1 | Yes | No | Gaze | Distance | Multiplicative | Comparative | 0.72 | 0.81 |
| G | 1 | Yes | No | Gaze | Constant | Multiplicative | Comparative | 0.51 | 0.30 |
| H | 1 | Yes | Yes | Constant | Distance | Multiplicative | Independent | 0.46 | 0.13 |
| I | 1 | Yes | No | Constant | Gaze | Multiplicative | Comparative | 0.47 | 0.56 |
| J | 1 | Yes | No | Gaze | None | Multiplicative | Comparative | 0.43 | 0.49 |

The hybrid variant used an alternative-wise gaze-discount and could thus accurately capture the relationship between relative dwell advantage and choice (Fig 5E and 5F). Again, it predicted an overall higher probability of target choices than GLA (Fig 5E), and this was primarily driven by the hybrid variant's predictions for individuals with strong attraction effects (S14 Fig) There was no meaningful difference between the two models in compromise trials (Fig 5F). These findings were also observed using relative fixation counts instead of relative dwell time (S15 Fig).

## Discussion

In this study, we investigated whether risky choice behaviour could be characterized by a gaze-dependent evidence accumulation framework, especially when choices are influenced by the context of available alternatives. In line with previous findings, we found choice behaviour to be context-dependent [49], but also subject to large interindividual differences. Importantly, participants' gaze behaviour was also modulated by the context of available alternatives, allowing a simple gaze-dependent evidence accumulation model derived from prior work on binary risky choice [1] to provide the best description of their choices. Finally, in a systematic search across a large space of possible model variants, we showed that gaze-dependent accumulation describes all participants' behaviour best. Predicting data from participants with particularly

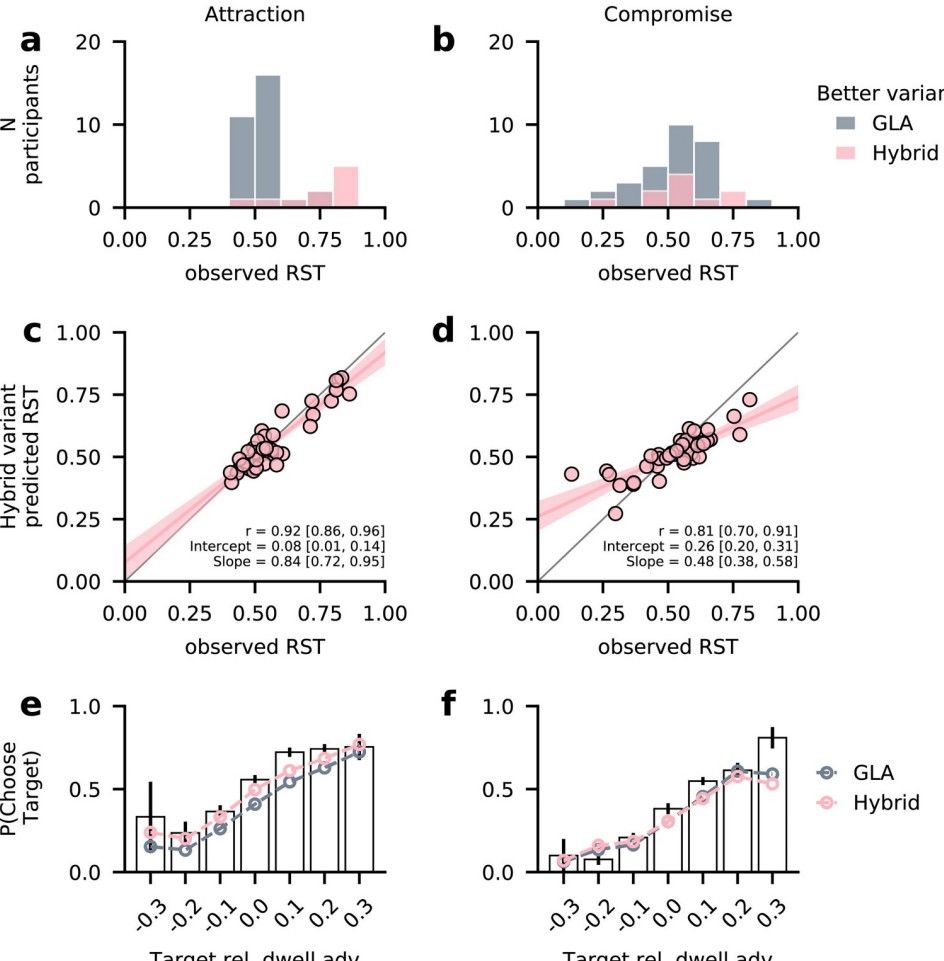

**Fig 5. Hybrid model variant details. (a, b)** Number of participants better described by the hybrid variant (pink) or the GLA (grey), dependent on strength of attraction **(a)** and compromise **(b)** effects. Participants with strong attraction effects were better described by the hybrid variant. **(c, d)** Individual observed and predicted RST in attraction **(c)** and compromise trials **(d)**. Compared to GLA (Fig 3I and 3J), the hybrid model better predicted strong attraction effects for some participants. Predictions of compromise effects are similar. **(e, f)** Observed and model-predicted probability of choosing the target alternative, depending on the target's relative dwell time advantage. Like other gaze-dependent models (Fig 3), the hybrid variant generally captured the positive association between gaze and choice. In contrast to GLA, however, it predicted an overall higher probability of choosing the target in attraction trials **(e)**. Predictions in compromise trials **(f)** are similar to GLA. White bars and error bars indicate mean ± s.e. observed data from even-numbered trials. Model predictions are based on 50 simulations for each odd-numbered trial.

strong attraction effects, however, required inclusion of an additional similarity-based inhibition mechanism.

Prior work could already demonstrate that choices between risky prospects can be influenced by the set of available alternatives, producing attraction [18,21–23], similarity [20], and other decoy effects [22]. These findings show that risky prospects, described by their winning probability and outcome, are subject to the same context-dependent influences as other multi-attribute stimuli, even though the natural (or normative) relationship between their attributes is multiplicative and not additive.

Our findings add to this literature, replicating the attraction effect and suggesting the presence of compromise effect in risky choice. As in prior studies of context effects [36,37], however, group-level evidence for the compromise effect was weaker than that for the attraction

effect. Prior explanations for the absence of context effects focused on attribute concreteness, that is, how clearly decision makers can extract the choice alternatives' objective attributes [50,51]. In our task, however, attributes were displayed with high concreteness (i.e., numerically), suggesting other moderating factors. For example, our indifference estimation procedure might not have resulted in full pairwise and transitive indifference between alternatives, possibly resulting in choices driven more by differences in fundamental subjective value than being modulated by context through gaze.

While most individuals showed standard compromise effects, the effect was also reversed for some. Such reversals have recently been linked to the information search process [52]. Importantly, we found the gaze-dependent model to capture these individual differences, relying on the context-dependent allocation of gaze in compromise trials.

We also find substantial individual differences in the strength of the attraction effect, ranging from almost no to large effects. The observed pattern includes a subgroup of participants predominantly choosing the target in attraction trials, that is also present in previous work in inference [35] and risky choice tasks [21]. Importantly, we could not find any evidence that these participants used simplifying choice rules based on the dominance relationship. The observed variability further emphasizes the importance of focusing on individuals' behaviour instead of group aggregates, if the goal is to understand the processes underlying individuals' choices [53].

Our study involved choices within sets of three risky gambles designed to elicit context effects. Prior research makes contradicting predictions about the direction of information search in this scenario: In the context of risky choice, empirical studies find a tendency towards within-alternative processing, disagreeing with non-compensatory, heuristic approaches and providing better support for compensatory strategies that assume integration of outcomes and probabilities [54–56] (but see refs. [57,58], or in the related domain of intertemporal choice: ref. [59]). In the domain of context effects, however, Noguchi & Stewart found pairwise comparisons between alternatives on single attribute dimensions to be the dominant pattern of information search [29]. They argue that these comparisons should form the basis of choice models describing context effects. Similarly, Marini *et al.* found that adding an asymmetrically dominated decoy to a choice set shifts eye movements towards the target's dominant attribute, and results in more transitions between target and decoy [30]. Cataldo & Cohen showed that the way information is displayed can influence the size and direction of context effects [60]: Alternative-wise presentation yielded similarity effects, whereas attribute-wise presentation, thought to induce comparisons between alternatives on single attributes, produced attraction and compromise effects. In line with the risky choice and context effects literature, we found participants to shift their gaze both within and between alternatives. While this does not constitute strong evidence for any particular process, this finding is compatible with current models of gaze-dependent accumulation in risky choice [1,3] and the GLA that assume within-alternative integration of probabilities and outcomes, and gaze-dependent accumulation and comparison processes to reach a decision.

Across decision making domains longer gaze towards an alternative is generally associated with a higher probability of choosing it [3–6,8,10,11,43,45,61–63]. This association is also present in choices between risky prospects [1,3,54,64]. While these results are mostly correlational, multiple studies found that manipulation of gaze towards an alternative increases its likelihood of being chosen, suggesting that gaze allocation influences choice [7–10,65,66].

The positive association between gaze and choice is also present in our data: Chosen gambles were looked at longer than others, and the effect increased over the course of a trial (see Supplementary Information). In addition, the probability of choosing an alternative increased with increasing relative dwell time. Gaze-dependent accumulation provides a formal account

of the association between choice and gaze data, as unattended alternatives' accumulation is diminished, making them less likely to be chosen. Conversely, if context effects were present in participants' gaze, this would enable such models to predict context-dependent choice. Our data illustrate that this contextual modulation of gaze is indeed present. While our results do not provide a full *explanation* of context-dependent choice behaviour (see below), they still suggest visual attention as a potential mediator.

Note that, in principle, GLA could produce a choice bias towards the alternative fixated last, by combining strong leakage with a strong gaze-discount: With a strong leak, predicted choices are influenced mainly by the information acquired in the final fixation. A strong gaze-discount could then bias choice towards the fixated alternative. The obtained parameter estimates, however, suggest only moderate strengths of leak and discounting, indicating that the model's good performance was not purely driven by effects of last fixations, which are often directed to the chosen alternative.

Our results are closely related to recent work showing that another behavioural effect in multi-alternative, multi-attribute choice is mediated by visual attention: Addition of a third alternative to a choice set has been shown to affect choice accuracy through value-based attentional capture in choices between risky prospects [2] and food items [43]. This mediation through gaze, formalized by a gaze-dependent accumulation model, provided a better description of the observed data than competing accounts. Adding to other work implicating mechanisms of visual attention in the emergence of context effects [60,67], our work shows how gaze can mediate context effects in a similar way: Choice sets affect the distribution of gaze, which in turn affects the choice process.

Many traditional models of risky decision making assume that one scale value is assigned to each alternative independent of the presence of others, and that choice probabilities are directly derived from these values (e.g., Luce's choice model [17]). These "simple scalability" theories include the most influential models of risky decision making (e.g., Expected Utility Theory [14]; and Prospect Theory [15,16]). They obey rational axioms of choice like IIA and therefore cannot account for context effects by design [24]. To explain context effects, multiple competing accounts have been proposed [22,25–27,68–71] (see ref. [44], for a taxonomy of mechanisms and overview).

These models often assume that an alternative's value is computed in comparisons to other alternatives on single attributes [25,27], that the considered attribute dimension switches stochastically from moment to moment [25,27,68], and that choices result from accumulation (often imperfect, i.e., leaky) of evidence until a threshold is reached [25–27,68]. Switching between attribute dimensions can introduce correlations between accumulators for similar alternatives, generating similarity effects [25,27,44,68]. To account for other context effects, these models employ additional mechanisms: for example, loss aversion, that is, differential weighting of advantageous and disadvantageous comparisons can produce attraction and compromise effects [27]. Distance-dependent inhibition between alternatives can yield similar results, by inhibiting similar alternatives more strongly and bolstering alternatives that are similar but dominant [25]. Non-linear value functions discounting alternatives with extreme attribute values can produce compromise effects [26].

However, while they propose precise psychological processes leading up to decisions, their relationship to observable process data, like eye movement recordings, remains implicit. For example, the switching between attribute dimensions is often considered an attentional mechanism [25,27,44,68], yet it is assumed to occur at every time-step (e.g., millisecond), and therefore cannot be mapped to observable eye-movement data without additional assumptions. Notably, thus far models of context-dependent choice do not include any gaze-dependency in

the decision process. This is in contrast to gaze-dependent accumulation models [1,5,6,11–13], which propose a formal account of the association between gaze and choice.

In our study, context-dependent choices were best described by a straightforward three-alternative extension of a gaze-dependent accumulation model that was previously applied to binary risky choices [1]. This model assumes that each alternative can be assigned a value by multiplicative integration of probability and outcome attributes, independently of other alternatives. Unlike simple scalable theories, however, it accumulates these values in a gaze-dependent fashion until a choice is made. Through its gaze-dependence, this model was able to predict individual differences in context effects. Yet, our analyses also revealed patterns challenging most basic gaze-dependent models and prompting further model development: for example, while greater dwell times towards the decoy were associated with more target choices, the GLA predicted increased decoy choices instead.

Still, the simple gaze-dependent and alternative-wise model performed best even across a large space of models, which included variants using additive attribute integration, attribute-wise gaze-discount and accumulation of comparative values. Such variants resemble extant models of context-dependent choice (e.g., MDFT), as they accumulate results from single attribute comparisons. Yet they performed worse, even when they included gaze-dependency. Our results thus question whether models of context-dependent choice must use attribute-wise comparisons over alternative-wise integration of attributes. However, we also found that strong context effects could be predicted best using an additional inhibition mechanism based on alternatives' similarity (which is comparative in nature), while still using alternative-wise valuation at its core, suggesting multiple parallel processes (i.e., within-alternative valuation *and* comparative mechanisms).

One limitation to the approach taken by the GLA is that it relies on observed fixation data to predict choices and context effects: While it *predicts* context effects, it does not *explain* why fixations are allocated in a context-dependent fashion (i.e., more towards target alternatives in our experiment). Addressing this question might again involve concepts and mechanisms used by other established models of context-dependent choice. For example, eye tracking studies on context effects showed that the *similarity* between alternatives (and which MDFT uses to scale feedback signals) is strongly associated with their fixation frequency [29,30]. Notably, focusing on the allocation of visual attention to understand context effects also opens a connection to research that identified other drivers of visual attention like visual saliency, position, size, and value in decision making [11,43,62,72–74]. Our results add the configuration alternatives' attributes to this list of stimulus features influencing gaze allocation.

A related limitation is that the gaze-dependent models presented here are not fitted to or make predictions about the response times at which decisions are made. This is also due to the sequence of fixation data (which also determines the response time) being used as *input* to the gaze-dependent models. Yet, response times are a meaningful of dimension of choice behaviour and are repeatedly associated with differences in context-effect strengths [38,40,51]. Therefore, more work is needed to analyse response time predictions of gaze-dependent models. A promising but technically and theoretically daunting way to address these limitations is the inclusion of generative models of the fixation process into gaze-dependent decision-making models. This approach was successful in some specific choice settings [2,6,11,43,62], with two recent studies investigating the optimal policy of attention allocation given a gaze-dependent decision mechanism and finding empirical data to match the optimal policy in simple two- and three-alternative choice [75,76]. Extensions to settings with multiple visually presented alternatives and attributes, however, remain a challenge to be addressed in future work.

Generally, our results suggest that extant models of context-dependent choice are likely to benefit from implementing gaze-dependence. Even further, explicitly formalizing the relationship

between model variables and eye movements yields testable predictions that can help distinguish and evaluate competing theories about the role of attention in context-dependent risky choice. The identified class of models is compatible with observed transition data, quantitatively captures the association of dwell time and choice probability, and uses the contextual modulation of gaze in addition to a distance-dependent inhibition mechanism to predict context effects.

## Methods

### Ethics statement

Written informed consent was obtained from all participants prior to the experiment. The experimental procedures were approved by Freie Universität's ethics committee.

### Participants

We determined a target sample size of 40 participants based on prior related work [1,32,54,56,75]. In total, we recruited 44 participants for the experiment. Four participants were excluded from the analyses: one due to a computer crash, two due to eye tracking calibration worse than 1.0° of visual angle, and one because of misunderstanding task instructions. All participants were required to have normal or corrected to normal vision with soft contact lenses. Participants relying on glasses or hard contact lenses were excluded from participation to ensure good eye tracking quality. The remaining 40 participants (25 female, 15 male) had mean ± s.d. age 27.2 ± 4.7. All participants received a base compensation of 8 Euros per hour and could win an additional bonus based on their choices during the experiment (see below).

### Task and stimuli

Participants performed a value-based choice task with stimuli designed to elicit attraction and compromise effects (Fig 1). Each trial started with a 1.5 s fixation cross at the screen centre. Then, three all-or-nothing gambles were presented next to each other on the screen. Gambles were described by a probability $p$ to win outcome $m$ (and winning nothing otherwise). Each gamble was enclosed by a rectangle. Gamble attributes $p$ and $m$ were arranged so that the vertical distance between two attributes of one option was equal to the horizontal distance between the centres of neighbouring alternatives. This distance was set to approximately 10.0° of visual angle. Alternative positions and attribute positions within each gamble were random in each trial. Participants were instructed to indicate their preference for one of the three gambles using their right hand and the keyboard's arrow keys. There was no time limit. After their choice, participants received a brief (0.3 s) feedback about their choice (but not about a gamble outcome).

Participants were instructed that after completing the task, one of the trials would be chosen randomly and the gamble chosen in this task would be played out for real money with a virtual wheel of fortune, using a later to be disclosed payment multiplier. This multiplier was set at 0.5 to scale winning bonuses to Freie Universität's payment standards.

Participants first performed three pairs of estimation and experiment blocks. Estimation consisted of a maximum of 30 trials with choices between two alternatives. These blocks served the purpose of determining individual indifference points for stimuli with varying levels of winning probability $p$ in an adaptive and integrated fashion. Participants were asked to indicate their preference between a fixed reference gamble $B$ ($p_B$ = 65%, $m_B$ = 15 EUR) and less risky option $A$ ($p_A$ = 75%, $m_A$ = 10 EUR). The outcome $m_A$ for option $A$ is then either increased (if $B$ was preferred) or decreased (if $A$ was preferred) and the procedure repeated. Indifference points for options $C_A$ and $C_B$ with probabilities $p_{C_A}$ = 85% and $p_{C_B}$ = 55% were

determined interleaved and in the same fashion. A single estimation block yielded a stimulus set with four options $A$, $B$, $C_A$ and $C_B$ designed in a way that option pairs $A−B$, $A−C_A$ and $B−C_B$ were approximately equally preferred and their distance in outcome $m_i$ was not less than 2 EUR. Additionally, asymmetrically dominated range-frequency decoy options $D_A$ and $D_B$ were introduced and designed to be 2% and 1 EUR worse than options $A$ and $B$, respectively.

The following experimental blocks had 75 ternary choice trials each, 32 of which were compromise trials balanced with respect to the target option (i.e., 16 choice sets $\{A, B, C_A\}$, 16 choice sets $\{A, B, C_B\}$), 32 attraction trials (16 choice sets $\{A, B, D_A\}$, 16 choice sets $\{A, B, D_B\}$) and 11 distractor trials showing randomly created options with expected value of 10 EUR and low (16–32%), medium (37–67%) and high (72–89%) probability $p$. To reduce repetition of stimulus values, for each trial, noise was added to each option's outcome $m$ (-1 EUR or no change) and probability $p$ (–3% or +3%). Asymmetrically dominated decoys $D_A$ and $D_B$ received the same noise as their focal option, to preserve dominance relation.

Participants performed 25 practice trials not relevant for their payout under supervision of the experimenter.

## Eye tracking

Participants' eye movements were recorded at 60 Hz using a table-mounted SMI Red eye tracker (SensoMotoric Instruments, Teltow, Germany). Participants were placed approximately 60 cm in front of the screen and instructed to minimize head movements during the task. Before each block, the eye tracker was calibrated using a 5-point calibration and validation procedure until a spatial resolution smaller than 1.0˚ visual angle was achieved horizontally and vertically. Participants were instructed to re-centre their gaze on the central fixation cross between trials.

Eye tracking data was pre-processed according to the following procedures: First, fixations, saccades and blinks were detected using SMI's Event-Detector software. Minimum fixation duration for detection was left at the default setting (80 ms). Blinks and saccades were discarded. Fixations were truncated when participants made a keyboard response. Next, rectangular areas of interest (AOIs) were constructed around the six screen locations that displayed stimulus attributes. Fixations towards non-AOI regions of the screen were discarded if they were preceded and followed by fixations to different AOIs. If they were preceded and followed by fixations towards the same AOI, the non-AOI fixation was re-coded to that AOI, too [6,11]. Finally, the total dwell time towards each alternative and attribute in each trial was computed by summing all fixation durations towards respective AOIs. Relative dwell time was computed by normalisation to the sum of all dwells in the trial.

## Behavioural modelling

**Baseline: Random choice.** The random choice model predicts equal choice probabilities $p = \frac{1}{3}$ for all three alternatives and serves as a benchmark against which other models can be compared.

**Prospect theory.** Prospect Theory (PT) [15] assumes that decision makers maximize subjectively weighted outcome utilities. Utilities of option outcomes $m_i$ are assumed to follow a power function with one free parameter $\alpha$:

$$U(m_i) = m_i^\alpha \tag{3}$$

Outcome probabilities are transformed into subjective decision weights using a probability weighting function [16]:

$$w(p) = \frac{p^{\gamma}}{(p^{\gamma} + (1-p)^{\gamma})^{\frac{1}{\gamma}}} \qquad (4)$$

where $\gamma$ ($0.28 \leq \gamma \leq 1$) is a free parameter controlling the shape of the weighting function. If $\gamma = 1$, the subjective weights equal the objective probabilities [77]. Predicted choice probabilities were computed using a soft-max choice rule [78] over weighted subjective utilities $x_i^{PT} = w(p_i) \cdot U(m_i)$:

$$p(x_i) = \frac{e^{\beta x_i}}{\sum_{j \in J} e^{\beta x_j}} \qquad (5)$$

Here, $J$ is the set of all available alternatives. The inverse temperature parameter $\beta$ controls the degree of randomness in the choice (choices become more deterministic with larger $\beta$).

**Multialternative decision field theory.** Multialternative Decision Field Theory (MDFT) [25] is a dynamic connectionist model for multi-attribute, multi-alternative choice. MDFT can predict similarity, attraction and compromise effects. The core principle of MDFT is the calculation of valences $V(t)$ at each point in time $t$. Valences are determined as

$$V(t) = CMW(t) + \varepsilon(t) \qquad (6)$$

where $M$ is a matrix containing all alternatives' attributes. $W(t)$ is a stochastic vector indicating the momentary weight distribution between attributes, according to a weight parameter $w$. $C$ is a contrast matrix, designed to perform attribute-wise contrasts between one option and the mean of other options' attributes. Finally, $\varepsilon(t)$ is a stochastic normally distributed noise component. Preferences $P(t)$ are generated integrating all valences $V(t)$ up to time $t$:

$$P(t) = SP(t-1) + V(t) \qquad (7)$$

Critically, $S$ is a square feedback matrix, thought to reflect the neurobiological mechanism of lateral inhibition between alternatives. The diagonal elements of $S$ determine how much the current preference state is influenced by the previous one, controlled by the decay parameter $\varphi_2$. Off-diagonal elements represent the feedback connections between alternatives. MDFT assumes that the connection strength between two alternatives depends on their perceived distance $D$ in attribute space, scaled by sensitivity parameter $\varphi_1$. Taken together, the feedback matrix $S$ is given by

$$S = I - \varphi_2 \exp(-\varphi_1 D^2). \qquad (8)$$

Here, $I$ is the identity matrix. $D$ is a matrix containing pairwise distances between alternatives. We used the distance function formalized in Hotaling *et al.* [79], where the distance between two alternatives is expressed in dominance- and indifference-directions and additionally scaled in dominance direction with an overweighting parameter $w_d$. The preference vector $P(t)$ is asymptotically normally distributed with mean $\xi$ and covariance matrix $\Omega$ [80], from which choice probabilities are derived as in the general Thurstone model [81].

In total, this MDFT implementation includes five free parameters: The attribute weight $w$, the decay parameter $\varphi_2$, the sensitivity parameter $\varphi_1$, the overweighting parameter $w_d$ and the variance $\sigma$ of the noise component $\varepsilon(t)$.

Typically, MDFT assumes an additive relationship between attributes. To accommodate for this, we log-transformed stimulus attributes probability $p$ and outcome $m$ [21]. Stimulus attributes were also rescaled to a range between 0 and 1 as in previous studies [32].

**Gaze-based models.** We included three models that use gaze data to predict choices: Two baseline models that ignore stimulus information and predict choices only based on gaze data, and one model adapted from previous work [1] that combines stimulus and gaze information in a leaky accumulation framework.

*Static gaze baseline model.* The first gaze-based model predicts choices from participants' cumulated dwell times towards each alternative. It assumes that preference strength $x_i^{GB_{stat}}$ for an alternative increases when it is fixated, irrespective of the its attributes $p_i$ and $m_i$:

$$x_i^{GB_{stat}} = d_i \tag{9}$$

where $d_i$ is the total dwell time (in s) towards alternative $i$ in a trial. Preference strengths $x_i^{GB_{stat}}$ are transformed into choice probabilities using the soft-max function (Eq (5)).

*Dynamic gaze baseline model.* The second gaze-based model uses the whole sequence of fixations in a trial to predict choices. It assumes that at each fixation, evidence in favour of the fixated alternative is accumulated, and that accumulated evidence is subject to leak. Formally,

$$X_i^{GB_{dyn}}(n) = \begin{cases} (1 - \lambda) \cdot X_i^{GB_{dyn}}(n - 1) + 1 & \text{if } i \text{ fixated} \\ (1 - \lambda) \cdot X_i^{GB_{dyn}}(n - 1) + 0 & \text{otherwise} \end{cases} \tag{10}$$

where all $X_i^{GB_{dyn}}(0) = 0$. The $\lambda$ parameter ($0 \leq \lambda \leq 1$) controls the strength of the accumulation leak. Choice probabilities are computed from the final accumulator values using the soft-max function (Eq (5)).

*Gaze-biased leaky accumulator model (GLA).* Finally, following a recent study on binary risky choice [1], we adapted a leaky accumulator model [82], where option values are discounted depending on eye movements as in the aDDM [6,11] and the related GLAM [12,13].

Here, the subjective utility for each alternative is computed as in PT, by first applying a probability weighting function (Eq (4)) [16], that transforms objective probabilities into subjective decision weights, and computing outcome utilities from a power function (Eq (3)). Subjective expected utilities are then given by $x_i^{GLA} = w(p_i) \cdot U(m_i)$.

Next, this model assumes that for each alternative subjective expected utilities are accumulated with leak and gaze bias over the time course of each trial. At each fixation $n$, accumulators evolve according to

$$X_i^{GLA}(n) = \begin{cases} (1 - \lambda) \cdot X_i(n - 1) + 1 \cdot x_i^{GLA} & \text{if } i \text{ fixated} \\ (1 - \lambda) \cdot X_i(n - 1) + \theta \cdot x_i^{GLA} & \text{otherwise} \end{cases} \tag{11}$$

where all $X_i(0) = 0$. The $\theta$ parameter ($0 \leq \theta \leq 1$) controls discounting of unattended alternative values. The $\lambda$ parameter ($0 \leq \lambda \leq 1$) controls the strength of the accumulation leak.

Predicted choice probabilities are again computed from the soft-max function (Eq (5)) over the final accumulator values $X_i^{GLA}$.

**Parameter estimation.** All models' parameters were estimated by minimizing the negative summed log-likelihood $-\ln(\hat{L})$ of observed choices under the model. Models were fit to the full set of choices, that is, across attraction, compromise, and distractor trials of each individual. Minimization was performed by a differential evolution algorithm [83] implemented in the SciPy Python library [84]. The algorithm was provided sensible *a priori* defined bounds

for each parameter and initialized randomly. Parameter bounds for MDFT were adapted from prior application of the model [32].

**Model comparison.** Individually best-fitting models were identified based on the Bayesian Information Criterion (BIC) [41], computed for each model $m$ as

$$BIC_m = -2\ln(\hat{L}) + \ln(n_{trials})k_m \tag{12}$$

Where $k_m$ is the number of free parameters of model $m$ and $\ln(\hat{L})$ is the summed log-likelihood of observed choices under model $m$. In addition, we computed each model's exceedance probability [42], denoting the probability of a model being implemented more frequently than others across the group.

## Model validation

To validate our parameter estimation and model selection procedures, we performed parameter- and model-recovery analyses. To test parameter recovery, we first estimated each model's parameters from the empirical data from each participant. Next, we generated synthetic data of identical size as the empirically observed data, by predicting choices for each empirical trial, using the obtained estimates. Finally, models were re-fit to the synthetic data, and known generating parameters were compared to the recovered estimates using Bayesian linear regression and correlation analyses [34,85].

To test model recovery, all models were fitted to the synthetic data generated from each of the models. Then, as in our model comparison, we performed Bayesian model selection [42] separately for each generating model.

## Switchboard analysis

We performed a switchboard analysis, similar to the one performed by Turner *et al.* [44] to further investigate which components of the cognitive model are particularly important in predicting the data. In a switchboard analysis, a cognitive model of the decision process is built, where individual model mechanisms can take different forms, or levels, which can be switched and combined with each other. One switch, or node, could for example be the integration of attributes to form item values. This integration could happen multiplicatively, so that expected outcomes are computed by multiplying outcome value and probability. It could also occur in a weighted additive fashion, so that both outcome value and probability make independent contributions to overall item value [48]. In the switchboard analysis, model variants using both implementations, and combinations with all other levels of other nodes, are constructed and fit to the behavioural data.

The switchboard analysis included different eye-movement related nodes, such as attribute and alternative wise gaze biases or gaze-dependent leakage and inhibition, so that the mechanisms that describe the data best can be identified and their relative contribution to model fit can be measured. All switchboard models resembled the general form of the gaze-dependent accumulation model presented in Glickman *et al.* [1] and the GLA adaptation to three items (see above and Fig 4A for a schematic). Here, evidence $X_i$ in favour of each item is accumulated over individual fixations. Accumulation can be subject to gaze-discount effects (so that non-fixated items accumulate less evidence), leak and inhibition over time. Choice probabilities are computed using a soft-max function (Eq (5)) over the final accumulator values. The general accumulation framework (in vector form, parallel for each item) is then given by

$$X(t) = S \times X(t-1) + \Theta C x \tag{13}$$

where $S$ is a square feedback matrix, combining the effects of accumulation decay (on its diagonal elements) and inhibition between accumulators (on off diagonal elements). $\Theta$ is the alternative-wise gaze discount vector (where the $i$th entry is set to 1 when item $i$ is fixated, and other entries are set to the discount value $\theta$, $0 \leq \theta \leq 1$). $C$ is a contrast matrix which, as in MDFT, can perform comparisons between the entries of the item value vector $x$. We now describe the different nodes and levels of the analysis, that are combined to generate the different model variants:

## Attribute integration

The attribute integration switch had two levels: First, outcome probability and outcome value could be integrated *multiplicatively*, so that expected outcome values per item are constructed. This level included the probability weighting function $w$ (Eq (4)) using a free parameter $\gamma$ ($0.28 \leq \gamma \leq 1$), and a utility function $U$ (Eq (3) free parameter $\alpha$ ($0 \leq \alpha$). The item values are given as

$$x_i = w(p_i)U(m_i).  \tag{14}$$

Alternatively, attribute integration could be implemented in a *weighted additive* fashion [48]. In this case, attributes were first normalized using divisive normalization [86] to make them commensurable on a single scale:

$$p_i^{norm} = \frac{p_i}{\sum_i^n p_i}  \tag{15}$$

and

$$m_i^{norm} = \frac{m_i}{\sum_i^n m_i}.  \tag{16}$$

Next, the normalized attributes would be combined *additively*, with weighting parameter $w_p$ ($0 \leq w_p \leq 1$), controlling the relative contribution of the probability attribute:

$$x_i = w_p p_i^{norm} + (1 - w_p)m_i^{norm}  \tag{17}$$

**Evidence comparison.**   The evidence comparison switch had two levels: First, item values $x_i$ could be accumulated *independently* for each alternative, without comparison to other alternatives. In this case the contrast matrix $C$ is set to an identity matrix. Second, item values $x_i$ could be accumulated in a *comparative* fashion. Then, the contrast matrix $C$ is set up to perform comparisons between each input $x_i$ and the mean of all other inputs $x_{j \neq i}$, as in MDFT. To this end, diagonal entries of $C$ are set to 1, and off-diagonal elements to $\frac{-1}{N-1}$, where $N$ is the number of alternatives (here $N = 3$).

**Alternative-wise gaze discount.**   This switch could take the values *"on"* or *"off"*. If switched on, the model included a free parameter $\theta$ ($0 \leq \theta \leq 1$) controlling the discount rate of unattended alternatives during accumulation. If switched off, the gaze discount vector $\Theta$ was set to one.

**Attribute-wise gaze discount.**   The analysis also included the option of attribute-wise gaze-dependent discounting (similar to the two-layer model from Glickman *et al.* [1] and the model presented in Fisher [46]). If switched on, stimulus attributes of the currently unattended dimension (e.g., probability, when a lottery outcome was fixated) were discounted by a free parameter $\eta$ ($0 \leq \eta \leq 1$). In combination with *additive* attribute integration, the attribute-wise gaze discount was applied after attribute normalization, but prior to the weighted addition.

For *multiplicative* attribute integration, attributes were discounted before entering probability-weighting and utility functions $w$ and $U$.

**Accumulation leak.**   We investigated three different forms of accumulation leak: First, accumulation without leak. In these variants, the diagonal elements of the feedback matrix $S$ were set to 1, resulting in no leak. The second possibility was uniform *constant* leak, where we estimated a parameter $\lambda$ ($0 \leq \lambda \leq 1$, where 1 indicates perfect memory without leak, and 0 indicates leak of all prior information), occupying the diagonal elements of the feedback matrix $S$. The third type of leak we investigated was *gaze-dependent*. Here, only accumulators of unattended alternatives leak according to the $\lambda$ parameter.

**Inhibition.**   We considered four types of inhibition between accumulators: First, independent accumulation without inhibition. In this case, all off-diagonal elements of $S$ were set to 0. Second, we considered uniform *constant* inhibition, where we estimated a parameter $\phi$ ($0 \leq \phi \leq 1$) and set each off-diagonal element in $S$ to $-\phi$, resulting in uniform inhibition (proportional to the accumulators' activation level), across items. Thirdly, we considered *distance dependent* inhibition, as implemented in MDFT (see Eq (8)). Here, the inhibition between accumulators is a function of the corresponding items' distance in attribute space. The distance is expressed in indifference and dominance directions, and the dominance direction is overweighted by a parameter $w_d$. As we did for MDFT, we log-transformed probability and outcome attributes and rescaled them to a range between 0 and 1 before applying the distance function. This implementation uses a sensitivity parameter $\phi$, a parameter estimating the relative importance of the probability attribute $w_p$ (this parameter is already used if attribute integration is *additive*), and the overweighting parameter of the dominance direction $w_d$. Note, we only computed off-diagonal elements of $S$ according to Eq (8), as the diagonal entries were controlled by the accumulation leak switch. Finally, we considered *gaze-dependent* inhibition. Here, the rationale is that only the accumulator of the currently attended item exerts inhibition over all others. In this type of inhibition, all off-diagonal elements of $S$ in the column corresponding to the currently attended item are set to $-\phi$, and others are set to 0.

**Total number of variants and parameter estimation.**   Exhaustive combination of all switch levels yields 192 model variants. The effective number of uniquely identifiable models was, however, reduced to 160 because for some variants comparative and independent accumulation versions cannot be distinguished when choice probabilities are derived from a soft-max choice rule with a freely estimated inverse temperature parameter over final accumulator values. This is the case for variants with no or constant inhibition and leak. Each variant was fit individually to the data from each participant by maximum-likelihood estimation, using a Differential Evolution optimization algorithm (see above). Like the extant models, all switch-board variants were fit to the set of all trials, including attraction, compromise, and distractor trials. As the number of parameters differ between model variants, we computed the BIC for each model and participant to obtain a measure of model fit, corrected for model complexity.

The optimization algorithm failed to find a solution better than a random model for 108 of 6400 (1.69%) of estimation runs. Since all model variants used the soft-max choice rule (Eq (5)) and therefore could predict random choices by setting the inverse temperature parameter $\beta$ to 0, this indicates non-convergence of the optimization algorithm. All but one non-converged estimation run used distance-dependent inhibition. We set the maximum-likelihood estimates of the failed runs to that of the nested random model for all analyses.

## Statistical modelling

We used Bayesian estimation (BEST) [33,34] of differences, effect size $d$ and associated 95% highest posterior density intervals (HDI$_{95}$) for all reported pairwise comparisons. Reported

correlation coefficients and associated $HDI_{95}$ result from Bayesian correlation analyses [85]. Regression estimates and $HDI_{95}$ result from Bayesian regression analyses implemented in PyMC3 [87].

## Software

The task was programmed in MATLAB (The Mathworks Inc., USA) using the PsychTool-Box [88]. Data processing and analyses were done in Python with numpy [89], scipy [84] and pandas [90] libraries. Bayesian analyses were implemented in PyMC3 [87], mixed models used bambi [91]. Exceedance probabilities were computed in MATLAB using SPM12 [92]. Figures were created using matplotlib [93], seaborn [94] and python-ternary [95].

## Data and code availability

All raw and preprocessed data, and scripts to reproduce all reported processing, analyses and figures are available at https://github.com/moltaire/gda-context.

## Supporting information

**S1 Fig. Mean response time and context-effect strength.** Associations of individual mean response times and context effect strength (RST) in **(a)** attraction and **(b)** compromise trials. Statistical annotation denotes mean intercept and slope with associated $HDI_{95}$ of Bayesian linear regression analyses.
(TIFF)

**S2 Fig. Distribution of gaze over the course of the trial depending on stimulus characteristics.** Distribution of gaze over the course of the trial depending on stimulus characteristics. Each panel shows the average dwell time towards AOIs for a given stimulus feature (e.g., horizontal and vertical position) across five time-bins. Data is shown separately for attraction **(a-f)** and compromise **(g-l)** trials.
(TIFF)

**S3 Fig. Weight estimates from regression analyses on absolute dwell times.** We performed two mixed-effects regression analyses of dwell time towards each AOI onto stimulus properties: **(a, c)** Regressing the total dwell time towards an AOI over a full trial onto AOI column, row, attribute rank (best, middle or worst value on the attribute), two dummy predictors coding alternative, attribute (probability or outcome) and whether the AOI belonged to the subsequently chosen alternative. **(b, d)** Second, we binned dwell times in each trial into five time-bins and added an interaction term with time-bin for each predictor. The panels show the interaction term weights. Analyses were carried out separately for attraction **(a, b)** and compromise **(c, d)** trials. Regression models had random intercepts and slopes across participants. Points and intervals mark posterior mean estimates and associated $HDI_{95}$ (coloured green if the interval excluded 0).
(TIFF)

**S4 Fig. Simulation of interacting GLA gaze-discount and leak mechanisms. (a)** Model-predicted choice probabilities for an item depending on its relative fixation count and whether it was fixated last or not. Different colours refer to different strengths of accumulation leak λ. **(b)**. Like **a** but without gaze-discount. Predictions for different values of λ fully overlap. All predictions are based on mean empirically estimated model parameters (except for leak parameters λ and the discount parameter θ in **b**) of trials with three alternatives with equal expected subjective value and all possible sequences of 6 fixations. Horizontal dashed lines

represent chance level. See S2 Note for details on the simulation analysis.
(TIFF)

**S5 Fig. Association of individual mean relative dwell time towards decoys and context effect strength (RST).** Bayesian linear regressions and correlation analyses of individual mean relative dwell time towards decoy alternatives and context effect strength (RST) in attraction **(a)** and compromise trials **(b)**. *P(Slope > 0)* denotes the posterior probability that the regression slope is larger than zero.
(TIFF)

**S6 Fig. Parameter recovery results.** Each row corresponds to results for one model. Each panel shows the relationship between true data generating, and recovered parameter values for a single model parameter. Generating parameters were obtained from fitting the models to empirical data from each participant. Annotations report Bayesian correlation coefficient *r*, and the intercept $\beta_0$ and slope $\beta_1$ estimates of a Bayesian regression analysis with $HDI_{95}$ given in brackets. Perfect, unbiased recovery would show an intercept of 0 and a slope of 1, with all points on the diagonal.
(TIFF)

**S7 Fig. Model recovery results with Expected Utility Theory included.** The large panel depicts a confusion matrix summarizing results of a model recovery analysis. Each cell refers to the posterior model probability of a fitted model (column) for a given generating model (row). Perfect recovery would be given by only values of 1 on the diagonal. The smaller matrix depicts exceedance probabilities for data from each generating model (row). Data generated from Prospect Theory (PT) was falsely attributed to Expected Utility Theory (EU).
(TIFF)

**S8 Fig. Model recovery results *without* Expected Utility Theory.** The large panel depicts a confusion matrix summarizing results of a model recovery analysis. Each cell refers to the posterior model probability of a fitted model (column) for a given generating model (row). Perfect recovery would be given by only values of 1 on the diagonal. The smaller matrix depicts exceedance probabilities for data from each generating model (row).
(TIFF)

**S9 Fig. Model comparison and predictions using GLA variant based on fixation *durations*.** Analog to main text Fig 3.
(TIFF)

**S10 Fig. GLA maximum likelihood estimates and relationships between parameters.** $\alpha$ is the utility parameter. $\beta$ is the inverse temperature parameter of the choice rule (0 = random choice). $\gamma$ is the probability weighting parameter (1 = linear weighting). $\lambda$ is the leak parameter (0 = perfect memory, 1 = full leak of all previous information). $\theta$ is the gaze-discount parameter (1 = no gaze-discount, 0 = maximum gaze-discount).
(TIFF)

**S11 Fig. Model-predicted choice probabilities.** Each panel shows distributions of participant-level mean model-predicted choice probabilities for the target, competitor, decoy and ultimately chosen alternative. Predictions for attraction and compromise trials are displayed separately in the top **(a-g)** and bottom rows **(h-n)**. Predictions were computed using individual maximum likelihood estimates. The hybrid model **(g, n)** was derived from the switchboard analysis and combines an alternative-wise gaze-discount with a distance-dependent inhibition mechanism. Violin plots show kernel density estimates of distributions of individual values.

Box plots mark lower and upper quartiles and median. Whiskers extend from first and last datum within 1.5 times the interquartile range from lower and upper quartiles, respectively. Values outside this range are indicated by open circles.
(TIFF)

**S12 Fig. Relative model fits between GLA and MDFT in relation to RST.** Relative model fits of MDFT (indicated by BIC difference between GLA and MDFT) tended to be higher for participants with higher RST in attraction trials (left panel; slope = 0.04, $HDI_{95}$ = [-0.01, 0.09] increase in RST per 100 unit increase in BIC difference, 93.6% of posterior mass above 0), but not compromise trials (right panel), even though 7 out of 9 participants with attraction RST above 0.7 were better described by GLA overall (participants left of dashed vertical line).
(TIFF)

**S13 Fig. Counts of individual best fitting switches.** Most participants were best described by model variants that included multiplicative attribute integration, with alternative-wise gaze discount, no attribute-wise gaze discount, constant leakage and no inhibition.
(TIFF)

**S14 Fig. Observed and model-predicted association of dwell time advantage and choice for participants with weaker and strong attraction effects. (a-f)** Data and model predictions for participants with weaker attraction effects (RST < 0.7). **(g-l)** Data and model predictions for participants with strong attraction effects (RST > 0.7) Each column refers to one choice alternative: Target (first column; **a, d, g, j**); Competitor (second column; **b, e, h, k**); Decoy (third column; **c, f, i, l**). Rows refer to trials in attraction **(a-c, g-i)** and compromise trials **(d-f, j-l)**. White and grey bars and error bars show observed mean ± s.e. choice probabilities computed from even-numbered trials, for participants with weaker and stronger attraction effects, respectively. Coloured lines indicate model predictions derived from 50 simulations for each odd-numbered trial.
(TIFF)

**S15 Fig. Observed and model-predicted probability of choosing the target alternative, depending on the target's relative fixation count.** Analog to main text Fig 5E and 5F. Pink and gray lines refer to the Hybrid and GLA models, respectively.
(TIFF)

**S1 Note. Additional analyses of gaze behaviour.**
(DOCX)

**S2 Note. Interactions of GLA gaze-discount and leak mechanisms.**
(DOCX)

**S3 Note. Analyses of process data with respect to simplifying choice rules.**
(DOCX)

**S1 Table. Summary of GLA estimates.** $\alpha$ is the utility parameter. $\beta$ is the inverse temperature parameter of the choice rule (0 = random choice). $\gamma$ is the probability weighting parameter (1 = objective probability weighting). $\lambda$ is the leak parameter (0 = perfect memory, 1 = full leak of all previous information). $\theta$ is the gaze-discount parameter (1 = no gaze-discount, 0 = maximum gaze-discount).
(DOCX)

**S2 Table. Overview of the nodes and switch-levels used in the switchboard analysis.** Switch levels that depend on gaze-data are shaded blue. Note that due to the model fitting procedure,

where model predicted choice probabilities are derived from a soft-max function over the final accumulator values, the comparison switch levels independent and comparative are not distinguishable for a subset of the model space, reducing the total number of unique variants to 160. (DOCX)

**S3 Table. Overview of average best fitting model variants.** All ten model variants that fit the data best on average used some form of gaze-dependence (blue shaded cells), mostly an alternative-wise gaze discount. *"n.d."* denotes variants where comparison mechanisms were not distinguishable by the analysis.
(DOCX)

# Author Contributions

**Conceptualization:** Felix Molter, Scott A. Huettel, Peter N. C. Mohr.

**Data curation:** Felix Molter.

**Formal analysis:** Felix Molter.

**Funding acquisition:** Peter N. C. Mohr.

**Investigation:** Felix Molter.

**Methodology:** Felix Molter, Armin W. Thomas.

**Software:** Felix Molter.

**Supervision:** Hauke R. Heekeren, Peter N. C. Mohr.

**Visualization:** Felix Molter.

**Writing – original draft:** Felix Molter.

**Writing – review & editing:** Felix Molter, Armin W. Thomas, Scott A. Huettel, Hauke R. Heekeren, Peter N. C. Mohr.

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
