## [Decision Letter · Decision Letter 0]

30 Nov 2021

Dear Dr Molter, 

Thank you very much for submitting your manuscript "Gaze-dependent evidence accumulation predicts multi-alternative risky choice behaviour" for consideration at PLOS Computational Biology.

As with all papers reviewed by the journal, your manuscript was reviewed by members of the editorial board and by several independent reviewers. In light of the reviews (below this email), we would like to invite the resubmission of a significantly-revised version that takes into account the reviewers' comments.

We cannot make any decision about publication until we have seen the revised manuscript and your response to the reviewers' comments. Your revised manuscript is also likely to be sent to reviewers for further evaluation.

Sincerely,

Stefano Palminteri

Associate Editor

PLOS Computational Biology

Wolfgang Einhäuser

Deputy Editor

PLOS Computational Biology

Reviewer's Responses to Questions

**Comments to the Authors:**

Reviewer #1: The present work tackles the issue of decoy effects in risky decisions. Various proposals have attempted to explain these violations of independence of irrelevant alternatives (IIA) in multi-alternative scenarios, and recently it has been highlighted that attentional capture processes may be driving them. This work expands this line, presenting a detailed analysis of how visual attention could be involved in attraction and compromise effects. A new behavioral and eye-tracking experiment plus a systematic and comprehensive modeling approach are included in this paper, showing the centrality that gaze may be playing in the appearance of these biases.

I reckon this work may be of interest for the neuroeconomics and decision-making community, adding further support to this alternative view of decoy effects, in addition to being a relevant application of attentional accumulation models. The paper is well written and figures very helpful (special mention to Figure 4 for schematizing the switchboard analysis). I suggest minor clarifications to some points:

1. In the introduction you mention three effects (attraction, compromise, and similarity). The results in the paper are focused on attraction and compromise, however, there is not much elaboration on the similarity effect. Please complement or correct this.

2. Given that attention is the factor driving attraction and compromise effects, do you think it makes sense to look for a modulation of the decoy effect by the amount of time the decoy was fixated? It may be interesting to check directly from participant’s behavior, whether (relative) gaze to decoy at individual level modulates individual RST magnitude (i.e. participants that gaze more the decoy have higher RST?). Even more, do you think is it possible to see at a within-participant level that variations in gaze could affect the magnitude of the effect? (e.g. split participant trials for lower and higher decoy gaze time and check if the RST is higher for high decoy gaze time trials). Maybe the relationship between (theta, attention bias parameter) and RST between subjects may also be informative.

3. Are participants paying more attention (more relative gaze time) to the decoy in compromise vs attraction trials? Do you think this could be connected to the fact that evidence for the compromise effect was weaker than for attraction effect? I think more elaboration on why compromise effects were modest in this experiment could be added to the discussion. Can you comment on why participants chose the compromise decoy with the higher outcome (Cb) in higher proportion to low value (low risk) decoy (Ca), when it was expected them to be the same? I'm guessing different traces of attention could be expected for these two cases

4. Please include further details regarding the model fit: is the fitting considering attraction and compromise trials together or they are fitted separately? Are you fitting to odd-numbered trials and testing with even-numbered trials? What was the method employed to extract the fixations for the simulations? What’s the situation for the model fitted in switchboard? Did you fit the two conditions together? Is table 2 estimated from attraction trials only?

5. Was there a reason to not include a model like GLA but without the accumulation leak? If I understood well, GLA and GBdyn models include a decay term that does not depend on gaze, so it may be raised that gaze AND accumulation decay are equally relevant to capture the context effects. Your switchboard analysis is tackling this issue, but maybe it is worth emphasizing the necessary inclusion of gaze over decay to allow the decoy effects to be captured.

6. It’s very interesting how the hybrid model combination from the switchboard can explain participants with higher RSTs. To stress this point, maybe is a good idea to include the average RST for the participants in each model category in table 2.

Additional comments:

1. Line 728 : the title is repeated (“Gaze-biased leaky accumulator model (GLA)”)

2. Line 557: “Additional mechanisms: For example, loss aversion”, capital F typo.

3. It may be a good idea to include the meaning of the acronyms for the model names in figure 3 legend.

Reviewer #2: In this manuscript, Molter et al. examined whether gaze-dependent evidence accumulation could explain context dependent effects in decisions under risk. First, they successfully elicited the attraction and compromise effects in their sample. Second, they fitted a number of computational models to the data. Models that did not take eye movements into account (EU & MDFT) provided a poorer fit to the data than models that did. In particular, the model that provided the best fit was a Gaze-dependent Leaky Accumulator (GLA), which assumed a multiplicative leaky integration of subjective amounts and probabilities. Then, the authors used the switchboard analysis to identify the contribution of specific model components. They found that the best model variant was equivalent to the GLA (provided the best for 17 subjects out of 40). Notably, a hybrid variant that in addition to gaze-dependent evidence accumulation also included distance dependent inhibition and comparative accumulation (as in the MDFT), provided the best fit for 9 subjects (especially those who showed strong attraction effect).

In general, the article is clearly written and timely, the findings are innovative and show that in most cases, context effects can be explained using eye movements (without the need for specific mechanisms), and the statistical analyzes and computational models are rigorous and carefully executed. I have a few comments that I hope will help to improve the manuscript:

1. In the introduction the authors mention the 3 context effects (attraction, compromise and similarity), however they focused only on two of them (compromise and attraction). What was the motivation for that?

2. The authors examined two models that did not take eye movements into account (EU & MDFT). I think that they should also add the CPT.

3. a. Figure 3A shows the paradigm used in the study. In my opinion, the figure will be clearer if another panel, illustrating the compromise effect, will be added.

b. I think that the figure will also be clearer if the authors mention in the caption that the location of the amounts and probabilities were randomly alternate on each trial.

4. Glickman et al. fitted the GLA model using fixations as accumulation unit, and found that this provided a slightly better fit than using time intervals (e.g., 1ms) as accumulation unit. Did the authors attempt to make this comparison? In this context, do the findings presented in Fig. 5c&f remain the same if instead of relative dwell time, the DV is the relative number of fixations?

5. Can the authors provide an intuition why the compromise effect was weaker in trial type CB than in trial type CA?

6. Can the authors explain how the sample size was determined?

7. The MDFT model predicts that the magnitude of context effects should increase with deliberation time. Is it the case in the data? (especially for the subjects who showed a strong attraction effect).

Reviewer #3: In their paper, Molter and colleagues combine eye tracking with computational modeling to investigate two prominent context effects, the attraction and compromise effects, in risky choice. They find significant/credible evidence for the attraction effect but not for the compromise effect (though the trends are going in the predicted direction). In both situations, they also find that people look more at targets than competitors (and least at decoys). They go on to model the data in two steps: First they compare different established models that include or do not include eye movements to improve predictions, and find that a gaze-dependent LCA model (GLA) fits best. Then they use a “switchboard” approach to identify which mechanisms contribute most to predicting the data. This analysis again finds that the GLA fits best, but for a minority of participants with a strong attraction effect a model that assumes distance-dependent inhibition and comparative evidence comparison (much like Multi-alternative Decision Field Theory, MDFT) provides the best account.

Overall, the paper is well written and, in my view, relatively easy to follow, even though it covers and brings together several comparatively complicated topics (context effects in multi-alternative choice, sequential sampling models, eye tracking, advanced cognitive modeling). Some of the results may appear a bit weak (e.g., the absence of a compromise effect, and the fact that the switchboard analysis doesn’t really reveal that additional mechanisms beyond the GLA are critical to accommodate the behavior in the current task). However, it is more important to judge papers by the timeliness of their research question and the rigorousness of their methods, and I am very positive w.r.t. both of the aspects regarding the present paper. So far, there are only few papers that have looked at context effects with eye tracking (the paper cites two of them: Marini et al. and Noguchi & Stewart), and none of them have combined these topics with sophisticated state-of-the-art cognitive modeling techniques. Thus, I am quite enthusiastic about this paper but have some suggestions for corrections and additions that should be addressed in a revision.

Major comments:

1. In the end, the winning model of this paper is basically the GLA. However, it is obvious that this model doesn’t really explain context effects. It is compatible with them as long as eye movements are in line with them (i.e., more fixations on targets than competitors), but on a mechanistic level, there is no explanation. The GLA does not predict that there will be more target-fixations (it just picks them up if this is how the data looks), and it does not assume a mechanism such as distance-dependent inhibition that would be able to predict context effects in some other (gaze-independent) way. I think this issue should be made more explicit in the discussion of the paper to be more clear about the limitations of GLA.

Relatedly, it should also be avoided that the paper gives the impression that context effects are solely driven by eye movements. I don’t think the authors want to convey such a message (and this is probably why they added the analyses around MDFT and the hybrid model), but again it would be good to be explicit here.

2. A second limitation of this work (and the GLA, at least in the way it is used here) is that the question of response times (RT) is not addressed at all. Even though the models are dynamic (sequential sampling) models that would be able to predict RT, they are fitted to choices only. This is because the RT (and the eye movements) are used more in the form of inputs (information that helps the model to make better predictions) rather than as outputs / dependent variables. All of this is fine in the context of the present work, but it should be made explicit that the issue of predicting RT is not addressed here.

Relatedly, models such as expected utility theory and MDFT have a disadvantage over models such as GLA, simply because they cannot leverage potential information that RT/eye-tracking data may contain. Thus, the first model comparison in the paper is somewhat unbalanced. This becomes most evident in the sentence starting in line 313 (page 17) “However, MDFT and EU generally could not capture the empirical association of gaze and choice;[…]”. Of course, they could not, because they do not make use of eye-tracking data. Theoretically, they could do so only if there is some third (hidden) factor that strongly ties eye movements to some element that these models do take into account. Speaking of: the small positive relationship between dwell time and choice probability that EU and MDFT seem to predict (see Fig. 3 c,d,e,f) is probably due to people looking more at high-value options (i.e., value-based attention; cf. Gluth et al., 2018/2020). This should be mentioned (there seems to be some evidence for it in Suppl. Fig. 1).

3. Also concerning the discussion, I think the two above-mentioned studies that have looked at the interplay of eye movements and context effects (Marini et al.; Noguchi & Stewart) warrant a more elaborate discussion. This may even be addressed earlier (in the introduction) to make clear what the novel contribution of the current paper over and above this previous work is supposed to be. I would also be interested in whether the authors can replicate the effects reported by Marini et al. and discussed on page 28 (fixations are shifted towards the target’s dominant attribute, and there are more target-decoy transitions).

4. The authors mention (page 17) that the gaze-dependent models appear to predict choices of dominated decoys in attraction trials if they have been looked at a lot. I was wondering, whether it could help to rescale/normalize the inputs to the model, so that comparatively low probabilities/magnitudes enter the models as negative values (or values very close to 0). In that case, looking at the decoy’s attributes could pull one away from this option.

5. Given that GLA uses Prospect Theory-based values as inputs, it would be important to add Prospect Theory as a non-gaze comparison model in addition to EU and MDFT.

Minor comments (not ordered by relevance):

6. Figure 1: in the figure, there probability difference between the decoy and the target is 3%, but in the legend, it says that the difference was always 2%. Please adjust / make consistent (whatever is correct, the figure or the text).

7. (I think) I can understand why the authors focused on 2 of the “big 3” context effects and left out the similarity effect (cf. Spektor et al., 2021, Trends Cogn Sci, for some discussion on why the similarity effect is rather distinct from the other two). However, it would again be helpful to be explicit here: what went into this decision?

8. Lines 131/page 7: “showing randomly created options with expected value of 10 EUR”. Was the expected value really 10, so for instance, in case of 5% probability, the amount shown on the screen would be 200?

Also, I do not understand this notation: “(-3%, 0%, 3%)”. If this refers to the lower and upper end of a uniform distribution, what is the middle value “0%” meant to be?

9. Line 173/page 10: “did not exclude”. I’d suggest to change this double negative into “included”.

10. I would be curious to know what the authors’ thoughts are about the presence of a rather strong compromise effect in the eye movements but not in the behavior.

11. Line 349 / page 19: typo: “differed” rather than “different”

12. Figure 4c. It would be good to also highlight the Hybrid model, e.g., with a differently-colored frame.

13. Line 486/page 28: “providing novel evidence for the compromise effect in risky choice”. I would lower the tone here, because it sounds as if the study found significant/credible effects for it.

14. Participants: I always ask authors to justify their sample size, whether it’s a based on a formal power analysis, previous related work, or just experience.

15. Line 634/page 34: typo: “mi”; the i should be subscripted.

16. Line 43/page 3: “and others that apply explicitly to”. I’d suggest to change to “and others that are applied explicitly to”.

Signed,

Sebastian Gluth

**Have the authors made all data and (if applicable) computational code underlying the findings in their manuscript fully available?**

Reviewer #1: Yes

Reviewer #2: Yes

Reviewer #3: Yes

PLOS authors have the option to publish the peer review history of their article (what does this mean?). If published, this will include your full peer review and any attached files.

Reviewer #1: No

Reviewer #2: No

Reviewer #3: **Yes: **Sebastian Gluth
---

## [Decision Letter · Decision Letter 1]

6 Apr 2022

Dear Dr Molter, 

Thank you very much for submitting your manuscript "Gaze-dependent evidence accumulation predicts multi-alternative risky choice behaviour" for consideration at PLOS Computational Biology. As with all papers reviewed by the journal, your manuscript was reviewed by members of the editorial board and by several independent reviewers. The reviewers appreciated the attention to an important topic. Based on the reviews, we are likely to accept this manuscript for publication, providing that you modify the manuscript according to the review recommendations.

Sincerely,

Stefano Palminteri

Associate Editor

PLOS Computational Biology

Wolfgang Einhäuser

Deputy Editor

PLOS Computational Biology

[LINK]

The paper will be formally accepted once revised to take into account the remaining suggestions of the Reviewers.

Reviewer's Responses to Questions

**Comments to the Authors:**

Reviewer #1: I want to thank the authors for addressing my points. I am satisfied with their answers and modifications. I am making some final extra suggestions just to clarify some points in the latest version:

Line 758: Since it is included in the discussion that GLA model does not explain fixation allocation, it may be useful to also mention these recent papers that explore policy and optimality of gaze allocation during decisions: Jang, Sharma and Drugowitsch, eLife, 2021; and Callaway, Rangel, Griffiths, PlosCompBiol, 2021. Of special interest may be Callaway et al. since it also models trinary choices.

Line 685: it is not clear to me what is referring “this way” in that sentence.

Line 832: not sure what ± 0 means.

It is a great work. Thank you for considering me for this review.

Reviewer #2: The authors have well addressed my previous comments in the revision. A few more minor comments:

1. I think that the authors should mention in the text that for the lotteries which were used, the predictions of the PT & CPT are equivalent.

2. line 238 – "Finally, it remains possible, that participants’ preferences shifted towards higher outcomes after the estimation block, leading to stronger preferences for the CB decoy and reduced compromise effects in this condition.". This explanation seems quite plausible; Do the data of the other conditions (e.g., attraction trials/fillers) support it?

3. line 91 – et. al. should be replaced with et al (the period should be removed).

4. Supplementary figure 4 – I think it would help to indicate that the dashed horizontal line represents chance level, and to add a reference to Supplementary note 2 in the caption of the figure.

Reviewer #3: The authors have done an excellent job with addressing my suggestions for their original manuscript, so I am happy to recommendation of the paper in its current form.

Signed,

Sebastian Gluth

**Have the authors made all data and (if applicable) computational code underlying the findings in their manuscript fully available?**

Reviewer #1: Yes

Reviewer #2: Yes

Reviewer #3: None

PLOS authors have the option to publish the peer review history of their article (what does this mean?). If published, this will include your full peer review and any attached files.

Reviewer #1: No

Reviewer #2: No

Reviewer #3: **Yes: **Sebastian Gluth

Figure Files:

Data Requirements:

Reproducibility:

References:

---

## [Editor Report · Decision Letter 2]

7 Jun 2022

Dear Dr Molter,

We are pleased to inform you that your manuscript 'Gaze-dependent evidence accumulation predicts multi-alternative risky choice behaviour' has been provisionally accepted for publication in PLOS Computational Biology.

Best regards,

Stefano Palminteri

Associate Editor

PLOS Computational Biology

Wolfgang Einhäuser

Deputy Editor

PLOS Computational Biology

---

## [Editor Report · Acceptance letter]

30 Jun 2022

PCOMPBIOL-D-21-01896R2 

Gaze-dependent evidence accumulation predicts multi-alternative risky choice behaviour

Dear Dr Molter,

I am pleased to inform you that your manuscript has been formally accepted for publication in PLOS Computational Biology. Your manuscript is now with our production department and you will be notified of the publication date in due course.

With kind regards,

Zsofia Freund
